# Joint User Scheduling and Hybrid Beamforming Design for Massive MIMO LEO Satellite Multigroup Multicast Communication Systems

**DOI:** 10.3390/s22186858

**Published:** 2022-09-10

**Authors:** Yang Liu, Changqing Li, Jiong Li, Lu Feng

**Affiliations:** 1Graduate School, Space Engineering University, Beijing 101416, China; 2Space Information School, Space Engineering University, Beijing 101416, China

**Keywords:** LEO satellite communications, massive MIMO, multigroup multicast, user scheduling, hybrid beamforming, robust, joint design, energy efficiency

## Abstract

In the satellite multigroup multicast communication systems based on the DVB-S2X standard, due to the limitation of the DVB-S2X frame structure, user scheduling and beamforming design have become the focus of academic research. In this work, we take the massive multi-input multi-output (MIMO) low earth orbit (LEO) satellite communication system adopting the DVB-S2X standard as the research scenario, and the LEO satellite adopts a uniform planar array (UPA) based on the fully connected hybrid structure. We focus on the coupling design of user scheduling and beamforming; meanwhile, the scheme design takes the influence of residual Doppler shift and phase disturbance on channel errors into account. Under the constraints of total transmission power and quality of service (QoS), we study the robust joint user scheduling and hybrid beamforming design aimed at maximizing the energy efficiency (EE). For this problem, we first adopt the hierarchical clustering algorithm to group users. Then, the semidefinite programming (SDP) algorithm and the concave convex process (CCCP) framework are applied to tackle the optimization of user scheduling and hybrid beamforming design. To handle the rank-one matrix constraint, the penalty iteration algorithm is proposed. To balance the performance and complexity of the algorithm, the user preselected step is added before joint design. Finally, to obtain the digital beamforming matrix and the analog beamforming matrix in a hybrid beamformer, the alternative optimization algorithm based on the majorization-minimization framework (MM-AltOpt) is proposed. Numerical simulation results show that the EE of the proposed joint user scheduling and beamforming design algorithm is higher than that of the traditional decoupling design algorithms.

## 1. Introduction

In recent years, LEO satellite communication systems have played an increasingly important role in wireless communication networks [1]. However, facing the high-performance requirements of future wireless communication systems, such as higher spectrum efficiency (SE) and increased EE, the performance of LEO satellite communication systems needs to be improved [2]. Applying the massive MIMO technology to LEO satellites is a good choice, and with the advantage of 5G technology [3] and the spatial multiplexing principle of MIMO technology [4], the performance of LEO satellite communication systems can be further improved. Meanwhile, by using the high-precision multiple beams generated by the massive MIMO technology and aggressive full frequency reuse scheme among beams, the performance of the communication system can be greatly improved. However, the full frequency reuse scheme will cause severe inter-beam interference [5], and adopting the beamforming design at the LEO transmitter side can efficiently manage it. In addition, the super-frame structure of multibeam satellite communications standards such as DVB-S2X [6] needs to apply the same beamformer to multiple users that share the same frame. Therefore, the multigroup multicasting principle can be used in the beamformer design. In this paper, we concentrate on the massive MIMO LEO satellite communication system forward link multigroup multicasting beamforming scheme [7].

In the beamforming design of the massive MIMO LEO satellite multigroup multicast communication system, the following issues need to be considered:User scheduling: Due to that only a few users can be bound into a DVB-S2X frame, and there are a large number of active users in each multicast group, it is necessary to design the user scheduling algorithm. Meanwhile, it should be noted that the interference between scheduled users depends on the beamforming design, which in turn depends on the scheduled users in other beams. Therefore, user scheduling and beamforming design are coupled, and the joint design scheme of user scheduling and beamforming needs to be considered.Channel errors: The LEO satellite has high orbital speed, which will produce a large Doppler shift and result in the channel phase deviation [8]. Meanwhile, the factors such as distortion of high-frequency devices, expiration of the CSI and large propagation delay also can cause the channel phase disturbance. Therefore, it is difficult to obtain accurate channel state information (CSI) at the LEO satellite transmitter. Due to the existence of CSI errors, the designed beamforming vector does not match the actual CSI, resulting in the reduction in the receiving gain and signal to interference plus noise ratio (SINR) of the user terminal. Then, the QoS will not be guaranteed. Thus, it is of practical significance to study the robust user scheduling and beamforming design.EE optimization: Due to the limited energy load of LEO satellites, to prolong the service life of the LEO satellite and improve the stability of the LEO satellite communications system, under the consideration of green communications and economic benefits, we need to pay attention to the EE optimization [9].Beamforming scheme: In the beamforming architectures of the massive MIMO technology, although the digital beamforming design can significantly improve the SE, it would bring high hardware complexity and high power consumption. Although the hardware overhead of hybrid beamforming architecture based on full connection is slightly higher than that of the partial connection architecture, it can balance the hardware complexity and system performance, and has higher cost performance. Therefore, in this paper, we selected the hybrid beamforming technology based on the full connection structure [10].

## 2. Related Works and Main Contributions

### 2.1. Related Works

There is extensive literature regarding user scheduling and beamforming design in the wireless multigroup multicast communication system. In Ref. [11], based on the perfect CSI, taking the throughput maximization as the optimization objective the authors studied the precoding design of the multibeam satellite communication system, and proposed a decoupling scheme of the user scheduling and beamforming design. In Ref. [12], considering the influence of CSI errors and taking the minimizing transmission power as the optimization objective, the robust multigroup multicast transmission scheme of the multibeam satellite communication system was investigated, and the low complexity beamforming algorithm and the user grouping algorithm were proposed, but the length limit of the DVB-S2X frame was not considered. In Ref. [13], based on the perfect CSI and taking the maximizing SE as the optimization objective, the multigroup multicast transmission design scheme of the frame-based multibeam satellite communication system was investigated, and the authors proposed a joint design scheme of the user scheduling and beamforming. However, the influence of CSI errors was not considered, and the user grouping algorithm was simple. In Ref. [14], based on the perfect CSI, the authors studied the user scheduling problem of the multicast transmission in the high-throughput satellite communication system. The user scheduling was decoupled into intra-beam scheduling and inter-beam scheduling, and the correlation degree was calculated by using the equivalent CSI; therefore, the interaction of intra-beam and inter-beam scheduling cannot be fully considered. In Ref. [15], the authors studied the user scheduling problem of the multibeam satellite communication system but did not consider the inter-beam interference caused by scheduling. In Ref. [16], based on the perfect CSI, the authors studied the joint scheduling and beamforming design problem for multiuser MISO downlink, and the message-based user grouping and scheduling algorithm was mainly proposed, but the impact of user grouping on system performance was not fully considered. In Ref. [17], the user scheduling and hybrid beamforming design of the massive MIMO orthogonal frequency division multiple access (OFDMA) communication system was studied, in scheme design. First, a joint design algorithm of user scheduling and analog beamforming was proposed; then, the digital beamforming matrix was solved by the weighted minimum mean-square error (WMMSE) algorithm. In Ref. [18], the authors studied the design of user scheduling and subcarrier allocation in the downlink of a massive MIMO OFDMA communication system and proposed a hybrid beamforming scheme. First, based on the optimal solution of digital beamforming, the analog beamforming matrix was obtained by a singular value decomposition algorithm. Then, the authors proposed an algorithm to solve the digital beamforming matrix and its corresponding scheduling users.

Most of the above research took the geosynchronous earth orbit (GEO) satellite or the terrestrial cellular network as the research object, and less used the LEO satellite. In addition, the optimization objectives mainly focused on maximizing SE and minimizing transmission power, and less on the EE optimization. Meanwhile, the analysis of CSI errors was insufficient, as some research considered the CSI errors, but did not analyze the influence of the Doppler shift. In terms of the user scheduling, the common idea was to adopt the decoupling scheme of user scheduling and beamforming design, without fully considering the coupling relationship between the user scheduling and the beamforming design.

### 2.2. Main Contributions

Inspired by the above research, we focus on the downlink transmission design of the massive MIMO LEO satellite multigroup multicast communication system. In scheme design, comprehensively considering the influence of CSI errors caused by the residual Doppler shift and the phase disturbance, we mainly investigate the robust joint user scheduling and hybrid beamforming design to maximize the system EE. Meanwhile, we take the constraints of the transmission power and QoS into account. The main works are summarized as follows:We establish the downlink transmission system model and channel model of the massive MIMO LEO satellite multigroup multicast communication system and analyze the CSI errors.Based on the CSI, we adopt a low complexity hierarchical clustering algorithm based on the Ward connection method to group users, which can lay a foundation for the joint user scheduling and beamforming design.We establish the joint user scheduling and hybrid beamforming design problem model based on EE maximization, and binary variables are defined to represent whether the user is scheduled or not. Then, we transform the optimization problem into a Boolean fractional programming (BFP) problem, which is also a quadratic constraint quadratic programming (QCQP) form problem.For the BFP problem in QCQP form, we invoke the quadratic transformation algorithm to handle the fractional programming form problem in the objective function. Meanwhile, the SDP algorithm is invoked to convert the objective function in QCQP form into a concave function, and some nonconvex constraints can be converted into linear constraints. In addition, we adopt the relaxation and penalty algorithm to deal with the Boolean constraint. Then, the optimization problem is equivalently transformed into a difference of convex (DC) programming problem.For the DC programming problem, an iterative optimization algorithm based on the CCCP framework is proposed. For the rank-one matrix constraint introduced by the SDP algorithm, a penalty iterative algorithm is adopted.For the solution of the digital beamforming matrix and the analog beamforming matrix in the hybrid beamformer, the MM-AltOpt algorithm is proposed.

## 3. System Model and Problem Formulation

### 3.1. System Model

As shown in Figure 1, we focus on the downlink of the massive MIMO LEO satellite multigroup multicast communication system, and the LEO satellite uses a UPA, which is composed of $N=N_{x}\times N_{y}$ antennas and $L\left(L\leq N\right)$ RF links and covers L multicast groups and K active users, where K≥L,K≥N. Let the multicast group covered by the lth beam be $U_{l}$ and the number of users in this multicast group be $\left|U_{l}\right|$, assuming that the number of users that can be accommodated in each DVB-S2X frame is $U_{s}$. It should be noted that each user terminal belongs to only one multicast group, i.e., $U_{i}\cap U_{j}=\emptyset ,\forall i,j\in \left\{1,\ldots ,L\right\},i\neq j$. Meanwhile, we assume that the user terminal is equipped with a single antenna capable of data stream demodulation. According to the DVB-S2X standard, in a transmission slot, multiple users’ data in a multicast group are multiplexed into a specific forward error correction (FEC) codeword to provide services for more users. The service process is shown in Figure 2.

The received signal yk,l of the kth user in the lth multicast group can be expressed as:(1)yk,l=hk,lHFRFFBB[:,l]sl+∑j=1,j≠lLhk,jHFRFFBB[:,j]sl+nk,l,k∈1,…,Ul,l∈1,…,L,
where the first term in (1) represents the expected received signal of the kth user in the lth multicast group, the second term represents the interference of other multicast groups and the third term represents the additive Gaussian white noise; hk,l∈ℂN×1 represents the channel vector of the kth user in the lth multicast group, FBB∈ℂL×L represents the digital beamforming matrix, FBB[:,l]∈ℂL×1 represents the digital beamforming vector of the lth multicast group, and FRF∈ℂN×L represents the analog beamforming matrix, where each element of FRF should meet the unit modulus element [19], i.e., FRFi,j=1. In addition, sl represents the signal of the multicast group Ul, which meets the unit power constraint, i.e., Εsl2=1, nl∼CN(0,σ2) represents the additive Gaussian white noise, which is related to the Boltzmann constant κ, system bandwidth B and the noise temperature T.

For the convenience of analysis, we set F∈ℂN×L=FRFFBB=[f1,f2,…,fL] as the hybrid beamforming matrix, and fl∈ℂN×1=FRFFBB[:,l] is the hybrid beamforming vector of the lth multicast group. Therefore, (1) can be rewritten as:(2)yk,l=hk,lHflsl+∑j=1,j≠lLhk,jHfjsl+nk,l,k∈1,…,Ul,l∈1,…,L

Due to the high orbital speed of LEO satellites and the long transmission delay, it is difficult to obtain the precise instantaneous CSI. To cope with this problem, we adopt the statistical CSI, and the channel vector between the LEO satellite and the kth user in the lth multicast group at instant t and frequency f can be modeled as follows [20]:(3)hk,l(t,f)=∑p=1Pk,lak,l,pej2π(fd(k,l,p)t−fτk,l,p)×Vk,l,p,
where f denotes the carrier frequency, ak,l,p,fd(k,l,p),τk,l,p are the complex channel gain, Doppler shift and propagation delay, respectively, Pk,l denotes the number of propagation paths and Vk,l,p∈ℂN×1 is the UPA array response vector, which can be given by
(4)Vk,l,p=V(φk,l,px,φk,l,py)=vNxsinφk,l,pycosφk,l,px⊗vNycosφk,l,py,
(5)vNx(sinφk,l,pycosφk,l,px)=1Nx1,e−j2πdλsinφk,l,pycosφk,l,px,…,e−j2πdλNx−1sinφk,l,pycosφk,l,px,
(6)vNy(cosφk,l,py)=1Ny1,e−j2πdλcosφk,l,py,…,e−j2πdλNy−1cosφk,l,py,
where φk,l,px, φk,l,py represent azimuth angle and pitch angle associated with the propagation path p of the kth user in the lth multicast group, respectively, λ denotes the wavelength and d represents the spacing of antenna elements, the value of which is usually λ/2 [5].

Note that the LEO satellite communication system is usually operated under the line of sight (LOS) transmission, and the channel vector can be modeled using the widely accepted Rician distribution model as follows:(7)hk,l=h¯k,l+h˜k,l,
where h¯k,l=κk,lγk,lκk,l+1×Vk,l represents the LoS component, h˜k,l=γk,lκk,l+1×Vk,l,c×Vk,lH represents the multipath component, κk,l denotes the Rician factor, Vk,l,c∈ℂNut×1∼ℂℕ0,∑ represents Rician component, Tr(∑)=1 and γk,l represents the average channel power, which mainly includes the transmit antenna gain Gleo, receiver antenna gain Gut and link power loss. The link power loss is mainly caused by the free space path loss LPfs and the atmospheric absorption loss LPat. Therefore, the average channel power γk,l can be written as
(8)γk,l=Gleo[dB]+Gut[dB]−LPat[dB]−LPfs[dB],
where LPfs can be given by LPfs=20log10Dk,l+log10f+log104π/c, c is the speed of light, Dk,l represents the transmission distance, LPat is related to the carrier frequency, temperature Th, pressure Ph and humidity ρh, which can be given by LPat=∫huthatLPatf,Th,Ph,ρhdh, LPatf,Th,Ph,ρh is the loss per meter, hut is the user’s height and hat is the atmosphere thickness. The specific calculation method of LPat can be found in the literature [21].

For the convenience of analysis, it is assumed that the parameters in the channel vector $h_{k,l}$ are constant within coherence time and change over time in a certain ergodic process. In (3), the Doppler shift $f_{d(k,l,p)}$ and the propagation delay $\tau_{k,l,p}$ usually cause CSI errors. Next, we focus on analyzing the influence of propagation delay and Doppler shift on CSI errors.

***Doppler shift***: In the LEO satellite communication systems, the Doppler shift is usually large, which is mainly composed of the Doppler shift $f_{d(k,l,p)}^{leo}$ generated by the LEO satellite motion and the Doppler shift $f_{d(k,l,p)}^{ut}$ generated by the users’ motion [22]. Since the transmission between the LEO satellite and user terminals is mainly under LOS, $f_{d(k,l,p)}^{leo}$ of different transmission paths can be considered to be the same, and we omit the path index of $f_{d(k,l,p)}^{leo}$, i.e., ${\left\{f_{d(k,l,p)}^{leo}\right\}}_{1}^{P_{k}}=f_{d(k,l)}^{leo}$; $f_{d(k,l)}^{leo}$ can be calculated using the LEO satellite ephemeris information and the location information of user terminals, as shown in Figure 3.
(9)$$f_{d(k,l)}^{leo}=-\frac{f}{c}\times \frac{\omega^{leo}r_{e}r\sin\left(\phi_{t}-\phi_{t_{0}}\right)\mu \left(\theta_{\max}\right)}{\sqrt{r_{e}^{2}+r^{2}-2r_{e}r\cos\left(\phi_{t}-\phi_{t_{0}}\right)\mu \left(\theta_{\max}\right)}},$$
where $\mu \left(\theta_{\max}\right)=\cos\left[\cos^{-1}\left(\frac{r_{e}}{r}\cos\theta_{\max}\right)-\theta_{\max}\right]$, $r_{e}$ denotes the earth radius, r represents the distance between the LEO satellite track point and the earth center, $\left(\phi_{t}-\phi_{t_{0}}\right)$ represents the angular distance of the earth’s surface along the LEO satellite trajectory from instant t to instant $t_{0}$, ω represents the angular velocity of the LEO satellite and c represents the speed of light.

The value of $f_{d(k,l,p)}^{ut}$ in different transmission paths is different, which is mainly caused by the movement of user terminals and surrounding scatterers. The power spectrum of $f_{d(k,l,p)}^{ut}$ follows the Jakes power spectrum model, and the normalized power spectrum can be expressed as:(10)$$S(f_{d(k,l,p)}^{ut})=\frac{1}{2\pi f_{d(k,l,p),\max}^{ut}\sqrt{1-{\left(\frac{f_{d(k,l,p)}^{ut}}{f_{d(k,l,p),\max}^{ut}}\right)}^{2}}}$$

The large Doppler shift in LEO satellite communication systems can make it difficult to receive correctly and result in the degradation of communication performance. In application, to mitigate the impact of Doppler shift, the solution of estimation and compensation is usually adopted [23]. The Doppler shift estimation mainly includes two steps: coarse estimation and fine estimation, which can refer to the literature [24]. When we obtain the estimated value of Doppler shift, $f_{d}^{e}$, the Doppler shift compensation of size $f_{d}^{cps}=f_{d}^{e}$ can be implemented at the receivers. It should be noted that due to that the Doppler shift changes rapidly, in addition to the low SINR at the receivers and the limited pilot length in the DVB-S2X frame, the Doppler shift estimation is usually inaccurate, which can result in the incomplete compensation. Then, there would be the residual Doppler shift $f_{d}^{rsd}$, which can cause the sliding of channel phase. According to the Doppler shift estimation theory based the Cramer–Rao bound, the variance in the Doppler shift estimation can be expressed as the Cramer–Rao lower bound (CRLB) [25], i.e.,
(11)$$\sigma_{f_{d}^{e}}^{2}=CRLB(f)=\frac{1}{SNR}\frac{3}{2\pi^{2}T^{2}N(N^{2}-1)},$$
where N is the pilot length, T is the sampling time and SNR is the signal-to-noise ratio. According to the properties of variance, after Doppler shift compensation of size $f_{d}^{cps}$, the variance of residual Doppler shift $f_{d}^{rsd}$ is equal to the variance of $f_{d}^{e}$, i.e.,
(12)$$\sigma_{f_{d}^{rsd}}^{2}=\sigma_{f_{d}^{e}}^{2}=\frac{1}{SNR}\frac{3}{2\pi^{2}T^{2}N(N^{2}-1)},$$

The influence of residual Doppler shift on channel phase errors can be expressed as $\phi_{f_{d}^{rsd}}=2\pi f_{d}^{rsd}\Delta T$, where ΔT represents the sum of the downlink propagation delay and the duration of the DVB-S2X frame. Therefore, the variance of $\phi_{f_{d}^{rsd}}$ can be expressed as $\sigma_{\phi_{f_{d}^{rsd}}}^{2}=4\pi^{2}\sigma_{f_{d}^{rsd}}^{2}\Delta T^{2}=\frac{1}{SNR}\frac{6\Delta T^{2}}{T^{2}N(N^{2}-1)}$, and $\phi_{f_{d}^{rsd}}$ follows a real-valued Gaussian distribution with mean zero and variance $\sigma_{\phi_{f_{d}^{rsd}}}^{2}$, i.e., $\phi_{f_{d}^{rsd}}∼N\left(0,\sigma_{\phi_{f_{d}^{rsd}}}^{2}\right)$. The channel error vector caused by $\phi_{f_{d}^{rsd}}$ can be expressed as $v_{\phi_{f_{d}^{rsd}}}={\left[e^{j\phi_{f_{d}^{rsd},1}},e^{j\phi_{f_{d}^{rsd},2}},\ldots ,e^{j\phi_{f_{d}^{rsd},N}}\right]}^{T}$.

***Propagation Delay***: The orbital height of LEO satellites is about 300 km to 2000 km, and the long transmission distance can cause the larger propagation delay. Note that the influence of atmosphere on propagation delay mainly includes ionospheric delay and tropospheric delay [26]. When the signal passes through the ionosphere, due to the refraction effect of the electromagnetic wave, the propagation path and speed of the signal will change. Meanwhile, the ionospheric delay is irregular, which is difficult to describe with a physical model. When the signal passes through the troposphere, the propagation speed, direction and path of the signal will change, which can result in propagation delay. The tropospheric delay is related to air pressure, air humidity and satellite elevation. The commonly used tropospheric delay correction model is given in the literature [27,28]. The round-trip delay of the LEO satellite with an orbit altitude of 1200 km is about 20 ms. The long propagation delay will lead to the expiration of CSI, which can result in CSI errors, phase disturbance and other problems. To handle this problem, the delay compensation of size $\tau^{cps}=\beta \tau_{\min}+(1-\beta )\tau_{\max},\;(0\leq \beta \leq 1)$ is usually implemented at the receivers, and $\tau_{k,l}^{\min}=\min{\left\{\tau_{k,l,p}\right\}}_{1}^{P_{k,l}}$ and $\tau_{k,l}^{\max}=\max{\left\{\tau_{k,l,p}\right\}}_{1}^{P_{k,l}}$ represent the minimum propagation delay and the maximum propagation delay of the kth user in the lth multicast group, respectively.

However, due to that the atmospheric propagation delay is irregular, the transmission delay cannot be fully compensated. Therefore, the incomplete delay compensation, expired CSI and distortion of high-frequency devices would cause the channel phase disturbance [29]. Let the phase disturbance be $\phi_{\tau }$, which follows a real-valued Gaussian distribution with mean zero and variance $\sigma_{\phi_{\tau }}^{2}$, i.e., $\phi_{\tau }∼N\left(0,\sigma_{\phi_{\tau }}^{2}\right)$. The channel error vector caused by $\phi_{\tau }$ can be expressed as $v_{\phi_{\tau }}={\left[e^{j\phi_{\tau ,1}},e^{j\phi_{\tau 2}},\ldots ,e^{j\phi_{\tau ,N}}\right]}^{T}$. 

In conclusion, considering the influence of the residual Doppler shift and the phase disturbance, the relationship between the real channel vector $h_{k,l}$ and the estimated channel vector ${\hat{h}}_{k,l}$ can be expressed as:(13)$$h_{k,l}={\hat{h}}_{k,l}⊙v_{\phi_{f_{d,k,l}^{rsd}}}⊙v_{\phi_{\tau ,k,l}}=diag\left(diag\left({\hat{h}}_{k,l}\right)v_{\phi_{f_{d,k,l}^{rsd}}}\right)v_{\phi_{\tau_{k,l}}},$$
where ⊙ represents the Hadamard product. Let the channel phase of the kth user in the lth multicast group be $\theta_{k,l}={\left[\theta_{k,l,1},\theta_{k,l,2},\ldots ,\theta_{k,l,N}\right]}^{T}$, which satisfies the uniform distribution between 0∼2π. Then, the real channel phase with phase errors at instant $t_{1}$ is as follows:(14)$$\theta_{k,l}(t_{1})=\theta_{k,l}(t_{0})+\phi_{f_{d,k,l}^{rsd}}+\phi_{\tau_{k,l}}$$

### 3.2. Problem Formulation

#### 3.2.1. User Clustering

Before the joint user scheduling and hybrid beamforming design, it is necessary to group the active users within the coverage of the LEO satellite. Based on the CSI, the hierarchical clustering algorithm is adopted to group users [30]. As shown in Figure 4 and Figure 5, the hierarchical clustering algorithm adopts the bottom-up method, where each user initially forms a group, and then according to the similarity measurement function, the user groups which meet the similarity threshold constraint are combined until the desired number of groups is formed.

We adopt the similarity measurement function among multicast groups based on the Ward connection method, i.e.,
(15)$$d\left(i,j\right)=\sqrt{\frac{2n_{i}n_{j}}{n_{i}+n_{j}}}dist(h_{i}^{eq},h_{j}^{eq}),$$
where $n_{i}$, $n_{j}$ represent the number of users of the group i and the group j, respectively, $h_{i}^{eq}$, $h_{j}^{eq}$ represent the equivalent CSI of the group i and the group j, respectively, and $dist(h_{i}^{eq},h_{j}^{eq})$ represents the Euclidean distance between the vector $h_{i}^{eq}$ and the vector $h_{i}^{eq}$, i.e.,
(16)$$dist(h_{i}^{eq},h_{j}^{eq})=\left\|\frac{h_{i}^{eq}}{\left\|h_{i}^{eq}\right\|}-\frac{h_{j}^{eq}}{\left\|h_{j}^{eq}\right\|}\right\|$$

#### 3.2.2. System Rate

Affected by CSI errors, both the ergodic communication rate and the ergodic SINR do not admit explicit expressions. To handle this challenge, the statistical average method is adopted to model the SINR and the communication rate. Therefore, the SINR and the communication rate of the kth user in the lth multicast group can be expressed as:(17)$$Ε\left\{SINR_{k,l}\right\}\approx \frac{Ε\left\{{\left|h_{k,l}^{H}f_{l}\right|}^{2}\right\}}{Ε\left\{\sum_{j=1,j\neq l}^{L}{\left|h_{k,l}^{H}f_{j}\right|}^{2}\right\}+\sigma^{2}},$$
(18)$$R_{k,l}\approx B\log_{2}\left(1+\frac{Ε\left\{{\left|h_{k,l}^{H}f_{l}\right|}^{2}\right\}}{Ε\left\{\sum_{j=1,j\neq l}^{L}{\left|h_{k,l}^{H}f_{j}\right|}^{2}\right\}+\sigma^{2}}\right),$$
where B denotes system bandwidth. Equations (17) and (18) are approximations with closed form, the feasibility of which have been discussed in detail in Refs. [31,32].

#### 3.2.3. Problem Description

We take the system EE as the optimization objective, and the EE is defined as the ratio of the system communication rate to the total power consumption, which can be modeled as:(19)$$EE=\frac{\sum_{l=1}^{L}\min\left({\left\{R_{k,l}\right\}}_{k=1}^{\left|U_{l}\right|}\right)}{P_{total}}=\frac{B\sum_{l=1}^{L}\log_{2}\left(1+\min{\left\{\frac{{\left|h_{k,l}^{H}f_{l}\right|}^{2}}{\sum_{j=1,j\neq l}^{L}{\left|h_{k,l}^{H}f_{j}\right|}^{2}+\sigma^{2}}\right\}}_{k=1}^{U_{s}}\right)}{P_{t}+P_{0}},$$
where $P_{total}$ represents the total power consumption, $P_{t}=\sum_{l}^{L}\left|f_{l}f_{l}^{H}\right|$ denotes the transmission power of the LEO satellite, and $P_{0}$ denotes the inherent power consumption of the communication system.

Let the Boolean variable $\eta_{k,l}\in \left\{0,1\right\}$ indicate whether the kth user in the lth multicast group is served, $\eta_{k,l}=1$ and $\eta_{k,l}=0$ indicate that the user can be served and not served, respectively, and $\eta =\left[\eta_{1},\eta_{2},\ldots ,\eta_{L}\right]$, $\eta_{l}={\left[\eta_{1,l},\eta_{2,l},\ldots ,\eta_{\left|U_{l}\right|,L}\right]}^{T}$. In conclusion, under the constraints of the transmission power and QoS, the problem of maximizing system EE can be modeled as:(20)$$Q_{1}:\;\underset{\eta ,f_{l},{\mathrm{SINR}}_{l}^{\min}}{\max}EE=\frac{B\sum_{l=1}^{L}\log_{2}\left(1+SINR_{l}^{\min}\right)}{\sum_{l}^{L}{\left|f_{l}f_{l}^{H}\right|}^{2}+P_{0}},$$
(21)$$s.t.\;C_{1}:\eta_{k,l}\in \left\{0,1\right\},\forall k,l,$$
(22)$$\;C_{2}:SINR_{k,l}\geq \eta_{k,l}SINR_{l}^{\min},\forall k,l,$$
(23)$$\;C_{3}:SINR_{l}^{\min}\geq SINR_{0},\forall l,$$
(24)$$C_{4}:\sum_{k=1}^{\left(U_{l}\right)}\eta_{k,l}=U_{s},\forall l,$$
(25)$$C_{5}:\sum_{l=1}^{L}{\left|f_{l}f_{l}^{H}\right|}^{2}\leq P_{T},$$
where $\;SINR_{l}^{\min}$ represents the minimum SINR of the lth multicast group and constraint $C_{3}$ represents that $\;SINR_{l}^{\min}$ should be greater than the minimum SINR constraint $SINR_{0}$. Constraint $C_{4}$ limits the number of scheduled users in each multicast group to $U_{s}$.

## 4. Joint User Scheduling and Hybrid Beamforming Design for Maximizing EE

In this section, we focus on the robust joint user scheduling and hybrid beamforming design strategy to maximize system EE. To handle the QCQP form problem and nonconvexity in optimization problem $Q_{1}$, the SDP method is applied to make the optimization problem more tractable. Then, we transform the optimization problem $Q_{1}$ into a DC programming problem. To address the DC programming problem, we adopt the CCCP algorithm. Finally, a penalty iterative algorithm is adopted to handle the rank-one matrix constraint.

### 4.1. SDP Algorithm

It is worth noting that the objective function and constraints $C_{2}$ and $C_{5}$ in the problem $Q_{1}$ involve the quadratic form of the variable $f_{l}$, therefore, $Q_{1}$ is the QCQP form problem. To handle this problem, we invoke the SDP algorithm, a new variable $W_{l}≜f_{l}f_{l}^{H}$ is introduced, and the positive semidefinite matrix $W_{l}$ needs to meet the constraints of $W_{l}\underline{≻}0$ and $rank\left(W_{l}\right)=1$. Then, the problem $Q_{1}$ can be equivalent to:(26)$$Q_{2}:\;\underset{\eta ,W_{l},SINR_{l}^{\min}}{\max}EE=\frac{B\sum_{l=1}^{L}\log_{2}\left(1+SINR_{l}^{\min}\right)}{\sum_{l}^{L}Tr(W_{l})+P_{0}},$$
(27)$$s.t.\;C_{1},C_{2},\;C_{3},\;C_{4}\;in\;Q_{1},$$
(28)$$C_{5}:\sum_{l}^{L}Tr(W_{l})\leq P_{T},$$
(29)$$C_{6}:W_{l}\underline{≻}0,\;\forall l,$$
(30)$$C_{7}:rank\left(W_{l}\right)=1,\;\forall l,$$

Similarly, the SINR and the communication rate of the kth user in the lth multicast group can be equivalently converted to:(31)$$SINR_{k,l}\approx \frac{Ε\left\{Tr\left(H_{k,l}W_{l}\right)\right\}}{Ε\left\{\sum_{j=1,j\neq l}^{L}Tr\left(H_{k,l}W_{j}\right)\right\}+\sigma^{2}},$$
(32)$$\begin{array}{ll} R_{k,l} & =B\log_{2}\left(1+\frac{Ε\left\{Tr\left(H_{k,l}W_{l}\right)\right\}}{Ε\left\{\sum_{j=1,j\neq l}^{L}Tr\left(H_{k,l}W_{j}\right)\right\}+\sigma^{2}}\right)=B\log_{2}\left(1+\frac{Tr\left(Ε\left\{H_{k,l}W_{l}\right\}\right)}{\sum_{j=1,j\neq l}^{L}Tr\left(Ε\left\{H_{k,l}W_{j}\right\}\right)+\sigma^{2}}\right)\\ & =B\log_{2}\left(1+\frac{Tr\left({\bar{H}}_{k,l}W_{l}\right)}{\sum_{j=1,j\neq l}^{L}Tr\left({\bar{H}}_{k,l}W_{j}\right)+\sigma^{2}}\right) \end{array}$$
where $H_{k,l}\in C^{M\times M}$ is the instantaneous channel autocorrelation matrix of the kth user in the lth multicast group and ${\bar{H}}_{k,l}\in C^{M\times M}$ is the long-term channel autocorrelation matrix. The relationship between the two can be expressed as:(33)$${\bar{H}}_{k,l}=Ε\left\{H_{k,l}\right\}≜Ε\left\{{\hat{h}}_{k,l}{\hat{h}}_{k,l}^{H}\right\}=diag\left({\hat{h}}_{k,l}\right)P_{f_{d,k,l}^{rsd}}Q_{\tau_{k,l}}diag\left({\hat{h}}_{k,l}^{H}\right),$$
where $P_{f_{d,k,l}^{rsd}}$ and $Q_{\tau_{k,l}}$ can be expressed as follows:


(34)
$$\begin{array}{ll} P_{f_{d,k,l}^{rsd}} & =Ε\left\{v_{\phi_{f_{d,k,l}^{rsd}}}v_{\phi_{f_{d,k,l}^{rsd}}}^{H}\right\}\\ & =Ε\left\{{\left[e^{j\phi_{f_{d,k,l}^{rsd},1}},e^{j\phi_{f_{d,k,l}^{rsd},2}},\ldots ,e^{j\phi_{f_{d,k.l}^{rsd},M}}\right]}^{T}\left[e^{-j\phi_{f_{d,k,l}^{rsd},1}},e^{-j\phi_{f_{d,k,l}^{rsd},2}},\ldots ,e^{-j\phi_{f_{d,k.l}^{rsd},M}}\right]\right\}\\ & =Ε\left\{\begin{array}{ccc} 1 & \cdots & e^{j\phi_{f_{d,k,l}^{rsd},1}}e^{-j\phi_{f_{d,k.l}^{rsd},M}}\\ \vdots & \ddots & \vdots \\ e^{j\phi_{f_{d,k.l}^{rsd},M}}e^{-j\phi_{f_{d,k,l}^{rsd},1}} & \cdots & 1 \end{array}\right\}\\ & =\left\{\begin{array}{ccc} 1 & \cdots & Ε\left\{e^{j\phi_{f_{d,k,l}^{rsd},1}}e^{-j\phi_{f_{d,k.l}^{rsd},M}}\right\}\\ \vdots & \ddots & \vdots \\ Ε\left\{e^{j\phi_{f_{d,k.l}^{rsd},M}}e^{-j\phi_{f_{d,k,l}^{rsd},1}}\right\} & \cdots & 1 \end{array}\right\} \end{array}$$


In (34), the diagonal elements of $P_{f_{d,k,l}^{rsd}}$ are all 1, and the elements in row i and column j on the non-diagonal are $Ε\left\{e^{j\phi_{f_{d,k.l}^{rsd},i}}e^{-j\phi_{f_{d,k,l}^{rsd},j}}\right\}=Ε\left\{e^{j\phi_{f_{d,k.l}^{rsd},i}}\right\}Ε\left\{e^{-j\phi_{f_{d,k,l}^{rsd},j}}\right\}$, according to $\phi_{f_{d}^{rsd}}∼N\left(0,\sigma_{\phi_{f_{d}^{rsd}}}^{2}\right)$,


(35)
$$Ε\left\{e^{j\phi_{f_{d,k.l}^{rsd},i}}\right\}=\int_{-\infty }^{\infty }e^{j\phi_{f_{d,k.l}^{rsd},i}}\frac{1}{\sqrt{2\pi }\sigma_{\phi_{f_{d,k,l}^{rsd}}}}e^{-\frac{\phi_{f_{d,k,l}^{rsd}}^{2}}{2\sigma_{\phi_{f_{d}^{rsd}}}^{2}}}d\phi_{f_{d,k.l}^{rsd},i}\;\;=e^{-\frac{\sigma_{f_{d,k,l}^{rsd}}^{2}}{2}}\int_{-\infty }^{\infty }\frac{1}{\sqrt{2\pi }\sigma_{\phi_{f_{d,k,l}^{rsd}}}}e^{-\frac{{\left(\phi_{f_{d,k.l}^{rsd},i}-j\sigma_{\phi_{f_{d,k,l}^{rsd}}}\right)}^{2}}{2\sigma_{\phi_{f_{d}^{rsd}}}^{2}}}d\phi_{f_{d,k.l}^{rsd},i}$$


Similarly, $Ε\left\{e^{-j\phi_{f_{d,k,l}^{rsd},j}}\right\}=e^{-\frac{\sigma_{\phi_{f_{d}^{rsd}}}^{2}}{2}}$. Therefore, $Ε\left\{e^{j\phi_{f_{d,k.l}^{rsd},i}}e^{-j\phi_{f_{d,k,l}^{rsd},j}}\right\}=Ε\left\{e^{j\phi_{f_{d,k.l}^{rsd},i}}\right\}Ε\left\{e^{-j\phi_{f_{d,k,l}^{rsd},j}}\right\}=e^{-\sigma_{\phi_{f_{d}^{rsd}}}^{2}}$.


(36)
$$\begin{array}{ll} Q_{\tau_{k,l}} & =Ε\left\{v_{\phi_{\tau_{k,l}}}v_{\phi_{t_{k,l}}}^{H}\right\}\\ & =Ε\left\{{\left[e^{j\phi_{\tau_{k,l},1}},e^{j\phi_{\tau_{k,l}2}},\ldots ,e^{j\phi_{\tau_{k,l},M}}\right]}^{T}\left[e^{-j\phi_{\tau_{k,l},1}},e^{-j\phi_{\tau_{k,l}2}},\ldots ,e^{-j\phi_{\tau_{k,l},M}}\right]\right\}\\ & =Ε\left\{\begin{array}{ccc} 1 & \cdots & e^{j\phi_{\tau_{k,l},1}}e^{-j\phi_{\tau_{k,l},M}}\\ \vdots & \ddots & \vdots \\ e^{j\phi_{\tau_{k,l},M}}e^{-j\phi_{\tau_{k,l},1}} & \cdots & 1 \end{array}\right\}\\ & =\left\{\begin{array}{ccc} 1 & \cdots & Ε\left\{e^{j\phi_{\tau_{k,l},1}}e^{-j\phi_{\tau_{k,l},M}}\right\}\\ \vdots & \ddots & \vdots \\ Ε\left\{e^{j\phi_{\tau_{k,l},M}}e^{-j\phi_{\tau_{k,l},1}}\right\} & \cdots & 1 \end{array}\right\} \end{array}$$


In (36), the diagonal elements of $Q_{\tau_{k,l}}$ are all 1, and the elements in row i and column j on the non-diagonal are $Ε\left\{e^{j\phi_{\tau_{k,l},i}}e^{-j\phi_{\tau_{k,l},j}}\right\}=Ε\left\{e^{j\phi_{\tau_{k,l},i}}\right\}Ε\left\{e^{-j\phi_{\tau_{k,l},j}}\right\}$, according to $\phi_{\tau }∼N\left(0,\sigma_{\phi_{\tau }}^{2}\right)$,
(37)$$\begin{array}{ll} Ε\left\{e^{j\phi_{\tau_{k,l},i}}\right\} & =\int_{-\infty }^{\infty }e^{j\phi_{\tau_{k,l},i}}\frac{1}{\sqrt{2\pi }\sigma_{\phi_{\tau_{k,l}}}^{}}e^{-\frac{\phi_{{}_{\tau_{k,l},i}}^{2}}{2\sigma_{\phi_{\tau_{k,l}}}^{2}}}d\phi_{\tau_{k,l},i}\\ & =e^{-\frac{\sigma_{\phi_{\tau_{k,l}}}^{2}}{2}}\int_{-\infty }^{\infty }\frac{1}{\sqrt{2\pi }\sigma_{\phi_{\tau_{k,l}}}}e^{{}^{-\frac{{\left(\phi_{\tau_{k,l},i}-j\sigma_{\phi_{\tau_{k,l}}}\right)}^{2}}{2\sigma_{\phi_{\tau_{k,l}}}^{2}}}}d\phi_{\tau_{k,l},i}\\ & =e^{-\frac{\sigma_{\phi_{\tau_{k,l}}}^{2}}{2}} \end{array}$$

Similarly, $Ε\left\{e^{j\phi_{\tau_{k,l},i}}\right\}=e^{-\frac{\sigma_{\phi_{\tau_{k,l}}}^{2}}{2}}$. Therefore, $Ε\left\{e^{j\phi_{\tau_{k,l},i}}e^{-j\phi_{\tau_{k,l},i}}\right\}=Ε\left\{e^{j\phi_{\tau_{k,l},i}}\right\}Ε\left\{e^{-j\phi_{\tau_{k,l},i}}\right\}=e^{-\sigma_{\phi_{\tau_{k,l}}}^{2}}$.

### 4.2. DC Programming

Since the constraint $C_{1}$ is a Boolean constraint and the constraint $C_{2}$ is a nonconvex constraint, the problem $Q_{2}$ is a nonconvex and nonsmooth combinatorial optimization problem. To handle this challenge, we can transform the problem $Q_{2}$ into a DC programming problem [33]. Therefore, the relaxation variable $\zeta_{k,l}$ is introduced as the lower bound of the SINR of the kth user in the lth multicast group, $\zeta =[\zeta_{1},\zeta_{2},\ldots ,\zeta_{L}]$, $\zeta_{l}={\left[\zeta_{l,1},\zeta_{l,2},\ldots ,\zeta_{l,\left|U_{l}\right|}\right]}^{T}$. The problem $Q_{2}$ can be equivalently converted to:(38)$$Q_{3}:\;\underset{\eta ,W,SINR_{l}^{\min},\zeta }{\max}EE=\frac{B\sum_{l=1}^{L}\log_{2}\left(1+SINR_{l}^{\min}\right)}{\sum_{l=1}^{L}Tr(W_{l})+P_{0}},$$
(39)$$s.t.\;C_{1},\;C_{3},\;C_{4},\;C_{5}\;,C_{6}\;,C_{7}\;in\;Q_{1},$$
(40)$$\;C_{2}:SINR_{k,l}\geq \zeta_{k,l},\forall k,l,$$
(41)$$\;C_{8}:\zeta_{k,l}\geq \eta_{k,l}SINR_{l}^{\min},\forall k,l,$$
where the constraint $C_{2}$ can be equivalently converted to:(42)$$C_{2}\Rightarrow 1+SINR_{k,l}\geq 1+\zeta_{k,l},$$

To further express (42) in the form of DC programming, we introduce new function variables $\Gamma_{k,l}\left(W\right)$ and ${Ι}_{k,l}\left(W,\zeta_{k,l}\right)$:(43)$$\Gamma_{k,l}\left(W\right)=\sigma^{2}+\sum_{j=1.j\neq l}^{L}Tr\left(H_{k,l}W_{j}\right),$$
(44)$${Ι}_{k,l}\left(W,\zeta_{k,l}\right)=\frac{\sigma^{2}+\sum_{j=1}^{L}Tr\left(H_{k,l}W_{j}\right)}{1+\zeta_{k,l}},$$

Therefore, the constraint $C_{2}$ can be rewritten as:(45)$$C_{2}\Rightarrow \Gamma \left(W\right)-Ι\left(W,\zeta_{k,l}\right)\leq 0,$$

In (45), $\Gamma_{k,l}\left(W_{l}\right)$ is the affine function of W, ${Ι}_{k,l}\left(W,\zeta_{k,l}\right)$ is the concave function of W and $\zeta_{k,l}$. The transformed constraint $C_{2}$ is a typical DC constraint.

Similarly, the constraint $C_{8}$ can be equivalently converted into the following DC form:(46)$$C_{8}\Rightarrow 4\zeta_{k,l}+{\left(\eta_{k,l}-SINR_{l}^{\min}\right)}^{2}\geq {\left(\eta_{k,l}+SINR_{l}^{\min}\right)}^{2},$$

In (38), the objective function in the problem $Q_{3}$ is a fractional programming problem with the sum-of-ratios form. To handle this problem, we invoke the quadratic transformation algorithm [34] and convert the problem $Q_{3}$ into the following form:(47)Q4:maxη,W,SINRlmin,ζEE=2qB∑l=1Lϒl(SINRlmin)12−q2∑l=1LTr(Wl)+P0,
(48)$$s.t.\;C_{1},\;C_{2},C_{3},\;C_{4},\;C_{5}\;,C_{6}\;,C_{7},C_{8}\;in\;Q_{3},$$
where q is the introduced auxiliary variable, and ${ϒ}_{l}({\mathrm{SINR}}_{l}^{\min})$ is the introduced auxiliary function, which can be expressed as:(49)$${ϒ}_{l}(SINR_{l}^{\min})=\log_{2}\left(1+SINR_{l}^{\min}\right),$$
(50)$$q=\frac{\sqrt{\sum_{l=1}^{L}{ϒ}_{l}({\mathrm{SINR}}_{l}^{\min})}}{\sum_{l=1}^{L}Tr(W_{l})+P_{0}},$$

In addition, for the nonsmooth combinatorial optimization problem caused by constraint $C_{1}$, we invoke a relaxation and penalty algorithm. Firstly, we relax constraint $C_{1}$ into $C_{1}\Rightarrow 0\leq \eta_{k,l}\leq 1,\;\forall k,l$. Meanwhile, to avoid the non-duality of the solution of $\eta_{k,l}$ caused by the relaxation, the penalty term $P\left(\eta_{k,l}\right)=\eta_{k,l}\log\eta_{k,l}+\left(1-\eta_{k,l}\right)\log\left(1-\eta_{k,l}\right)$ is introduced into the objective function: let $\lambda_{1}>0$ be the penalty factor, and the problem $Q_{4}$ can be equivalently converted to:(51)Q5: maxη,W,SINRlmin,ζEE=2qB∑l=1Lϒl(SINRlmin)12−q2∑l=1LTr(Wl)+P0+λ1∑l=1LPηk,l,
(52)$$s.t.\;C_{2}:\Gamma_{k,l}\left(W\right)-{Ι}_{k,l}\left(W,\zeta_{k,l}\right)\leq 0,\forall k,l,$$
(53)$$C_{3},\;C_{4},\;C_{5}\;,C_{6}\;,C_{7}\;in\;Q_{4},$$
(54)$$C_{8}:4\zeta_{k,l}+{\left(\eta_{k,l}-SINR_{l}^{\min}\right)}^{2}\geq {\left(\eta_{k,l}+SINR_{l}^{\min}\right)}^{2}\forall k,l,$$

### 4.3. CCCP Algorithm

From (51), (52) and (54), it can be seen that the problem $Q_{5}$ is a DC programming problem. To handle this challenge, the CCCP framework algorithm is a common method to solve the DC programming problem [35], which is an iterative framework including two operations: convexification and optimization. In the convexification step, by adopting the first-order Taylor expansion, the convex part of the objective function and the concave part of the constraint function can be linearized; then, the DC programming problem is transformed into a convex problem. It should be noted that the convex problem obtained from the convexification step provides a global lower bound for the original problem, and the optimization step is mainly to maximize the lower bound. Meanwhile, the performance of the CCCP algorithm is closely related to the initial point of the variables, but the equality constraint $C_{4}$ of the problem $Q_{5}$ limits the selection of the initial point. To find a feasible initial point, we substitute the constraint $C_{4}$ into the objective function and set the penalty factor $\lambda_{2}>0$. Then, the problem $Q_{5}$ can be equivalently converted to:(55)Q6:maxη,W,SINRlmin,ζEE=2qB∑l=1Lϒl(SINRlmin)12−q2∑l=1LTr(Wl)+P0+λ1∑l=1LPηk,l−∑lLλ2∑kUlηk,l−Us2
(56)$$s.t.\;C_{2},C_{3},\;C_{5}\;,C_{6}\;,C_{7},C_{8}\;in\;Q_{5},$$

***Convexification***: Let ${\left(\eta_{k,l},SINR_{l}^{\min},W,\zeta \right)}^{\left(t-1\right)}$ be the estimated value of variables $\left(\eta_{k,l},SINR_{l}^{\min},W,\zeta \right)$ in iteration t−1 of the problem $Q_{6}$. In iteration t, for the convex part $\lambda_{1}\sum_{l=1}^{L}P\left(\eta_{k,l}\right)$, we adopt the first-order Taylor expansion to replace it, which is reflected in line three of Algorithm 1. The first-order Taylor expansion of $P\left(\eta_{k,l}\right)$ can be expressed as:(57)$$P{\left(\eta_{k,l}\right)}^{te}=P\left(\eta_{k,l}^{\left(t-1\right)}\right)+\left(\eta_{k,l}-\eta_{k,l}^{\left(t-1\right)}\right)\nabla P\left(\eta_{k,l}^{\left(t-1\right)}\right),$$
(58)$$\nabla P\left(\eta_{k,l}^{\left(t-1\right)}\right)=\log\left(\eta_{k,l}^{\left(t-1\right)}\right)-\log\left(1-\eta_{k,l}^{\left(t-1\right)}\right),$$

Similarly, in the constraint $C_{2}:\Gamma_{k,l}\left(W\right)-{Ι}_{k,l}\left(W,\zeta_{k,l}\right)\leq 0$, we replace the concave function ${Ι}_{k,l}\left(W,\zeta_{k,l}\right)$ with its first-order Taylor expansion, which is reflected in line three of Algorithm 1. The first-order Taylor expansion of ${Ι}_{k,l}\left(W,\zeta_{k,l}\right)$ can be expressed as:(59)$${Ι}_{k,l}{\left(W,\zeta_{k,l}\right)}^{te}={Ι}_{k,l}\left(W^{\left(t-1\right)},\zeta_{k,l}^{\left(t-1\right)}\right)+\nabla^{T}{Ι}_{k,l}\left(W^{\left(t-1\right)},\zeta_{k,l}^{\left(t-1\right)}\right)\left[\begin{array}{c} {\left\{W_{l}-W_{l}^{\left(t-1\right)}\right\}}_{l=1}^{L}\\ \zeta_{k,l}-\zeta_{k,l}^{\left(t-1\right)} \end{array}\right],$$
(60)$$\nabla {Ι}_{k,l}\left(W^{\left(t-1\right)},\zeta_{k,l}^{\left(t-1\right)}\right)={\left[{\left\{\frac{{\left({\bar{H}}_{k,l}\right)}^{T}}{1+\zeta_{k,l}^{\left(t-1\right)}}\right\}}_{l=1}^{L},-\frac{\sigma^{2}+\sum_{l=1}^{L}Tr\left({\bar{H}}_{k,l}W_{l}^{\left(t-1\right)}\right)}{{\left(1+\zeta_{k,l}^{\left(t-1\right)}\right)}^{2}}\right]}^{T},$$

In the constraint $C_{8}:4\zeta_{k,l}+{\left(\eta_{k,l}-SINR_{l}^{\min}\right)}^{2}\geq {\left(\eta_{k,l}+SINR_{l}^{\min}\right)}^{2}\forall k,l$, we replace the concave function ${\left(\eta_{k,l}-SINR_{l}^{\min}\right)}^{2}$ with its first-order Taylor expansion, which is reflected in line 3 of Algorithm 1. The first-order Taylor expansion of ${\left(\eta_{k,l}-SINR_{l}^{\min}\right)}^{2}$ can be expressed as:(61)$${\left(\eta_{k,l}-SINR_{l}^{\min}\right)}^{2,te}={\left(\eta_{k,l}^{\left(t-1\right)}-SINR_{l}^{\min,\left(t-1\right)}\right)}^{2}+{\left[\begin{array}{c} 2\left(\eta_{k,l}^{\left(t-1\right)}-\mathrm{SIN}R_{l}^{\min,\left(t-1\right)}\right)\\ -2\left(\eta_{k,l}^{\left(t-1\right)}-SINR_{l}^{\min,\left(t-1\right)}\right) \end{array}\right]}^{T}\left[\begin{array}{c} \eta_{k,l}-\eta_{k,l}^{\left(t-1\right)}\\ SINR_{l}^{\min}-SINR_{l}^{\min,\left(t-1\right)} \end{array}\right],$$

***Optimization***: The optimization step is reflected in line nine of Algorithm 1. According to (57), (59) and (61), the problem $Q_{6}$ can be equivalently converted to
(62)Q7:maxη,W,SINRlmin,ζEE=2qB∑l=1Lϒl(SINRlmin)12−q2∑l=1LTr(Wl)+P0+λ1∑l=1LPηk,lte−∑lLλ2∑kUlηk,l−Us2
(63)$$s.t.\;C_{3},\;C_{5}\;,C_{6}\;,C_{7}\;in\;Q_{6},$$
(64)$$C_{2}:\Gamma_{k,l}\left(W\right)-{Ι}_{k,l}{\left(W,\zeta_{k,l}\right)}^{te}\leq 0,\forall k,l,$$
(65)$$C_{8}:4\zeta_{k,l}+{\left(\eta_{k,l}-SINR_{l}^{\min}\right)}^{2,te}\geq {\left(\eta_{k,l}+SINR_{l}^{\min}\right)}^{2}\forall k,l,$$

For the problem $Q_{7}$, the variables ${\left(\eta_{k,l},SINR_{l}^{\min},W,\zeta \right)}^{\left(t+1\right)}$ can be updated by the iterative optimization.

***Feasible Initial Point***: It should be noted that the CCCP algorithm needs a feasible initial point to ensure that the algorithm converges to a stationary point, as the selection of the initial point can affect the performance of the CCCP algorithm. To find a better initial point, we adopt the following method, which is reflected in line one of Algorithm 1.Initialize $\eta_{k,l}^{\left(0\right)}\approx 0$, $SINR_{0}=1$;Find $W^{\left(0\right)}$, the following optimization problem are modeled:(66)$$P_{FES}:\;\left\{W^{\left(0\right)}\right\}:\;\underset{W}{\min}\;\sum_{l=1}^{L}Tr(W_{l}),$$
(67)$$s.t.\;C_{1}:\left\|\sigma^{2}\ldots Tr{\left({\bar{H}}_{k,l}W_{j}\right)}_{j\neq l}\ldots \right\|\leq \frac{Tr({\bar{H}}_{k,l}W_{l})}{\eta_{k,l}SINR_{0}},\forall k,l,$$
(68)$$C_{2}:\sum_{l=1}^{L}Tr\left(W_{l}\right)\leq P_{T},$$If $P_{FES}$ is feasible, proceed to the next step, otherwise, update $\eta_{k,l}^{\left(0\right)}=\delta \eta_{k,l}^{\left(0\right)},0<\delta <1$ and repeat step 2;Based on the $W^{\left(0\right)}$ obtained in step 2, calculate the SINR of each user, i.e., ${\mathrm{SINR}}_{k,l}^{\left(0\right)},\forall k,l$, and update $\eta_{k,l}^{\left(0\right)}$ according to $\eta_{k,l}^{\left(0\right)}=\min\left\{1,\frac{SINR_{k,l}^{\left(0\right)}}{SINR_{0}}\right\}$;Based on $\eta_{k,l}^{\left(0\right)}$ and $W^{\left(0\right)}$, calculate $\zeta^{\left(0\right)}$ and ${\left\{SINR_{l}^{\min,}{}^{\left(0\right)}\right\}}_{l=1}^{L}$.

### 4.4. Penalty Iteration Algorithm

It should be noted that the SDP algorithm brings the nonconvex and nonsmooth constraint, i.e., $C_{7}:rank(W_{l})=1$. To solve the rank-one constraint, many existing research directly relaxes the rank-one constraint in the optimization step [36], and then judges whether the optimization solution ${\left\{W_{l}\right\}}_{l=1}^{L}$ meets the rank-one constraint. If so, the eigenvalue decomposition (EVD) algorithm is directly adopted to obtain the hybrid beamforming vectors ${\left\{f_{l}\right\}}_{l=1}^{L}$ according to $W_{l}=f_{l}f_{l}^{H}$, and if the optimization solution ${\left\{W_{l}\right\}}_{l=1}^{L}$ does not meet the rank-one constraint, the Gaussian randomization algorithm (GRA) is usually adopted. The basic idea of the GRA is as follows: Firstly, a set of candidate Gaussian vectors ${\left\{{\left\{w_{g,l}\right\}}_{g=1}^{G}\right\}}_{l=1}^{L}$ are generated based on the optimization solution ${\left\{W_{l}\right\}}_{l=1}^{L}$, where G represents the number of the Gaussian randomization. Secondly, from the generated G-group candidate Gaussian vector pool, combined with the power redistribution among the multicast groups, a group of Gaussian vectors is selected as the optimal hybrid beamforming matrix to maximize the objective function in the problem $Q_{7}$. It should be noted that in the case of the high-dimensional matrix, GRA has high complexity and large performance loss, resulting in poor availability. To this end, we adopt a feasible algorithm with the better performance: the penalty iteration algorithm.

According to the properties of the matrix, $rank(W_{l})=1$ is equivalent to $Tr(W_{l})-\lambda_{\max}\left(W_{l}\right)=0$. Therefore, the nonsmooth method is adopted to transform the constraint $rank(W_{l})=1$ in the problem $Q_{7}$ into the following form:(69)$$C_{7}:Tr(W_{l})-\lambda_{\max}\left(W_{l}\right)\leq 0,$$
where $\lambda_{\max}\left(W_{l}\right)$ is the function of solving the maximum eigenvalue. It should be noted that for any positive semidefinite matrix $W_{l}\underline{≻}0$, the inequality $Tr(W_{l})-\lambda_{\max}\left(W_{l}\right)\geq 0$ is always true, which means that the transformed constraint $C_{7}$ and $Tr(W_{l})-\lambda_{\max}\left(W_{l}\right)=0$ are equivalent. Then, we can obtain that the matrix $W_{l}$ has only one non-zero eigenvalue and can be given by
(70)$$W_{l}=\lambda_{\max}\left(W_{l}\right)w_{l,\max}w_{l,\max}^{H},$$
where $w_{l,\max}$ is the corresponding unit eigenvector. Therefore, the problem $Q_{7}$ can be converted into
(71)Q8:maxη,W,SINRlmin,ζEE=2qB∑l=1Lϒl(SINRlmin)12−q2∑l=1LTr(Wl)+P0+λ1∑l=1LPηk,lte−∑lLλ2∑kUlηk,l−Us2
(72)$$s.t.\;C_{2},\;C_{3},\;C_{5}\;,C_{6}\;,C_{8}\;in\;Q_{7},$$
(73)$$C_{7}:Tr(W_{l})-\lambda_{\max}\left(W_{l}\right)\leq 0,$$

In the iterative calculation of the problem $Q_{8}$, based on the obtained $W_{l}$, if the value of $Tr(W_{l})-\lambda_{\max}\left(W_{l}\right)$ is small enough, the matrix $W_{l}$ can be considered to meet the rank-one constraint, which is reflected in line seven of Algorithm 1. Therefore, to make the value of $Tr(W_{l})-\lambda_{\max}\left(W_{l}\right)$ as small as possible, we adopt the penalty iteration algorithm and substitute the constraint $C_{7}$ into the objective function in the problem $Q_{8}$. Therefore, the problem $Q_{8}$ can be converted into
(74)$$Q_{9}:\underset{\eta ,W,SINR_{l}^{\min},\zeta }{\max}EE=F\left(\eta ,W,SINR_{l}^{\min},\zeta \right)-\lambda_{3}\sum_{l}^{L}\left(Tr\left(W_{l}\right)-\lambda_{\max}\left(W_{l}\right)\right),$$
(75)$$s.t.\;C_{2},\;C_{3},\;C_{5}\;,C_{6}\;,C_{8}\;in\;Q_{8}$$
where Fη,W,SINRlmin,ζ=2qB∑l=1Lϒl(SINRlmin)12−q2∑l=1LTr(Wl)+P0+λ1∑l=1LPηk,lte−∑lLλ2∑kUlηk,l−Us2 and $\lambda_{3}>0$ is the penalty factor, which is generally larger enough to ensure that a smaller value of $Tr(W_{l})-\lambda_{\max}\left(W_{l}\right)$ can be obtained. 

According to (74), the iterative calculation of the problem $Q_{9}$ can maximize function $F\left(\eta ,W,SINR_{l}^{\min},\zeta \right)$ and minimize function $Tr(W_{l})-\lambda_{\max}\left(W_{l}\right)$. It should be noted that $Tr(W_{l})$ is an affine function and $\lambda_{\max}\left(W_{l}\right)$ is nonsmooth, which can result in the nonsmoothness of the objective function in the problem $Q_{9}$. To handle the challenge, we replace $\lambda_{\max}\left(W_{l}\right)$ with its first-order Taylor expansion. The subgradient of $\lambda_{\max}\left(W_{l}\right)$ is $\frac{\partial \lambda_{\max}\left(W_{l}\right)}{\partial W_{l}}=w_{l,\max}w_{l,\max}^{H}$ and its first-order Taylor expansion can be expressed as follows, which is reflected in line eight of Algorithm 1.
(76)$$\lambda_{\max}\left(W_{l}^{\left(t\right)}\right)\geq \lambda_{\max}\left(W_{l}^{\left(t-1\right)}\right)+\langle w_{l,\max}w_{l,\max}^{H},W_{l}^{\left(t\right)}-W_{l}^{\left(t-1\right)}\rangle ,$$
where $\langle w_{l,\max}w_{l,\max}^{H},W_{l}^{\left(t\right)}-W_{l}^{\left(t-1\right)}\rangle =Tr\left({\left(w_{l,\max}w_{l,\max}^{H}\right)}^{H}\left(W_{l}^{\left(t\right)}-W_{l}^{\left(t-1\right)}\right)\right)$.

We substitute (76) into the objective function in the problem $Q_{9}$ to replace $\lambda_{\max}\left(W_{l}\right)$, and the problem $Q_{9}$ can be expressed as:(77)$$Q_{10}:\underset{\eta ,W,SINR_{l}^{\min},\zeta }{\max}EE=F\left(\eta ,W,SINR_{l}^{\min},\zeta \right)-\lambda_{3}\sum_{l}^{L}\left(Tr\left(W_{l}\right)-\lambda_{\max}\left(W_{l}^{\left(t-1\right)}\right)+\langle w_{l,\max}w_{l,\max}^{H},W_{l}^{\left(t\right)}-W_{l}^{\left(t-1\right)}\rangle \right),$$
(78)$$s.t.\;C_{2},\;C_{3},\;C_{5}\;,C_{6}\;,C_{8}\;in\;Q_{9},$$

In conclusion, the robust joint user scheduling and hybrid beamforming design algorithm for the massive MIMO LEO satellite multigroup multicast communication system is shown in Algorithm 1.
**Algorithm 1:** Joint user scheduling and hybrid beamforming design algorithm.**Input:** CCCP algorithm iteration index k, thresholds ε1, penalty iteration algorithm iteration index m, thresholds ε2, penalty factor λ1, λ2, λ3.1. Initial: η,W,SINRlmin,ζk=0, qk=0.2. while EEk−EEk−1≥ε13.  Convexification step by (57), (59), (61).4.  Calculation qk, substitute qk into (77).5.  Optimization step.6.  Let η,W,SINRlmin,ζm=0=η,W,SINRlmin,ζk=0.7.  while TrWlm−λmaxWlml=1L≥ε28.   Calculate the maximum eigenvalue λmaxWl of Wlm and the corresponding eigenvector wl,maxm.9.   Using CVX toolbox, calculate the variables η,W,SINRlmin,ζoptm at the mth iteration according to (77).10.    If Wlm+1lL≈WlmlL, then11.     Update λ3=2λ3.12.   else13.     Update m=m+1.14.   end15.  end16.  Update η,W,SINRlmin,ζk+1=η,W,SINRlmin,ζm, k=k+1, λ1=λ1+1, λ2=λ2+1.17. end**Output**:η,W,SINRlmin,ζopt.

## 5. Convergence and Complexity Analysis

### 5.1. Convergence

The effectiveness of the algorithm depends on its convergence. For the convergence of the CCCP algorithm, the convergence has been proven by [37]. To prove the convergence of the penalty iteration algorithm, let the variable solution and objective function value of the optimization problem $Q_{10}$ be ${\left(\eta ,W,SINR_{l}^{\min},\zeta \right)}^{\left(k+1\right)}$ and $F_{EE}\left({\left(\eta ,W,SINR_{l}^{\min},\zeta \right)}^{\left(k+1\right)}\right)$ at the kth iteration. Therefore, the convergence can be proved as follows:


(79)
$$\begin{array}{l} F_{EE}\left({\left(\eta ,W,SINR_{l}^{\min},\zeta \right)}^{\left(k+1\right)}\right)=F\left({\left(\eta ,W,SINR_{l}^{\min},\zeta \right)}^{\left(k+1\right)}\right)-\lambda_{3}\sum_{l}^{L}\left(Tr\left(W_{l}^{\left(k+1\right)}\right)-\lambda_{\max}\left(W_{l}^{\left(k+1\right)}\right)\right)\\ \geq F\left({\left(\eta ,W,SINR_{l}^{\min},\zeta \right)}^{\left(k+1\right)}\right)-\lambda_{3}\sum_{l}^{L}\left(Tr\left(W_{l}^{\left(k+1\right)}\right)-\lambda_{\max}\left(W_{l}^{\left(k\right)}\right)-\langle w_{l,\max}w_{l,\max}^{H},W_{l}^{\left(k+1\right)}-W_{l}^{\left(k\right)}\rangle \right)\\ \overset{by\left(75\right)}{\geq }F\left({\left(\eta ,W,SINR_{l}^{\min},\zeta \right)}^{\left(k\right)}\right)-\lambda_{3}\sum_{l}^{L}\left(Tr\left(W_{l}^{\left(k\right)}\right)-\lambda_{\max}\left(W_{l}^{\left(k\right)}\right)\right)\\ =F_{EE}\left({\left(\eta ,W,SINR_{l}^{\min},\zeta \right)}^{\left(k\right)}\right) \end{array}$$


The convergence can be proved according to (79). Therefore, after initializing the values of ${\left(\eta ,W,SINR_{l}^{\min},\zeta \right)}^{\left(k=0\right)}$, $\lambda_{1}^{\left(k=0\right)}$, $\lambda_{2}^{\left(k=0\right)}$ and $\lambda_{3}^{\left(k=0\right)}$, the proposed algorithm can iteratively converge to an optimal solution by setting a reasonable convergence threshold.

### 5.2. Complexity

The complexity of the algorithm directly affects its performance. In the algorithms adopted, the complexity of the hierarchical clustering algorithm can be calculated according to the connection algorithm, similarity measurement criteria and hierarchical grouping process, and the algorithm complexity can be expressed as $O\left(LK^{2}N\right)$. The complexity of the joint user scheduling and hybrid beamforming design algorithm is closely related to the number of multicast groups and scheduling users. In addition, the number of optimization variables and constraints in the CCCP algorithm and the penalty iteration algorithm can also affect the complexity [38]. In the problem $Q_{10}$, the number of optimization variables is $2\sum_{l=1}^{L}\left|U_{l}\right|+2L$, the number of convex constraints is $\sum_{l=1}^{L}\left|U_{l}\right|$ and the number of linear constraints is $\sum_{l=1}^{L}\left|U_{l}\right|+3L$. Let the number of iterations in the penalty iteration algorithm and the CCCP algorithm be $I_{p}$ and $I_{c}$, respectively. In conclusion, the overall complexity of the proposed algorithm is $O\left(LK^{2}N+I_{p}I_{c}\left(2\sum_{l=1}^{L}\left|U_{l}\right|+2L\right)\left(2\sum_{l=1}^{L}\left|U_{l}\right|+3L\right)\right)$.

According to the algorithm complexity, the proposed joint user scheduling and hybrid beamforming design algorithm has a strong timeliness in small dimensional communication systems. However, for large dimensional communication systems, such as the satellite communication system, the number of active users is usually large. According to the complexity analysis, with the increase in the total number of active users, the convergence speed of the algorithm would gradually slow down, and the complexity would gradually increase. Considering the characteristics of the LEO satellite communication system, the delay caused by the high complexity is unacceptable, which would affect the overall performance of the communication system. To handle this problem, considering the balance of the algorithm performance and complexity, before the joint user scheduling and hybrid beamforming design, we can appropriately reduce the system dimension by adding the user preselection step in the algorithm process. The user preselection step can reduce the total number of active users in each transmission, and then we carry out the joint user scheduling and hybrid beamforming design for preselected users.

## 6. User Preselection Algorithm

In the user preselection step, $U_{l,p}$ users in the lth multicast group are preselected as the user representatives, where $U_{s}<U_{l,p}\leq \left|U_{l}\right|$. The user preselection process is shown in Figure 6. The symbols (circles, squares and triangles) represent the users in different the multicast group. The red circles represent preselected users and scheduling users.

The selection of preselected users can affect the performance of the joint user scheduling and hybrid beamforming design algorithm, which depends on the preselected algorithm. In the beamforming design of the multigroup multicast communication system, the beamforming vector is oriented to multiple users in the multicast group. Therefore, in the process of user preselection, to maximize the receive gain of each user, i.e., $\left|h_{k,l}^{H}f_{l}\right|$, the beamforming vector $f_{l}$ of the lth multicast group should be collinear with the users’ channel vectors in the multicast group as far as possible. Therefore, the channel vectors of the preselected users in the same multicast group should also be strongly linearly correlated. Meanwhile, the interference among multicast groups should also be taken into account in the user preselection stage. To reduce the interference among multicast groups, the channel vectors of preselected users among different multicast groups should be orthogonal. Similarly, the beamforming vector of the multicast group should be orthogonal to the users’ channel vectors in other multicast groups.

In conclusion, we adopt a low complexity user preselection algorithm, which can preselect orthogonal users among the different multicast groups and linearly correlated users in the same multicast group. The proposed algorithm is divided into two steps, as follows:The first step: according to the orthogonal criterion [11], a user is preselected for each multicast group in turn, which is reflected in line three and line four of **Step 1** in Algorithm 2;The second step: based on the users of each multicast group selected in the first step, linearly correlated users are selected for each multicast group, which is reflected in line two and line three of **Step 2** in Algorithm 2.

The specific preselection process of the two steps is as follows:
**Algorithm 2:** User preselection algorithm.**Step 1: Orthogonal user preselection algorithm among the different multicast groups.****Input**: CSI.1. Let Id1=Indexmaxhk,l,∀k,l, select the user with the largest channel gain, Id1 is the index of the user.2. while l≤L,l≠Id13.   For all users in the lth multicast group, calculate Zk,l=hk,lIN−∑j=Id1IdlhjHhjhj22 in turn.4.  Idl=IndexmaxZk,l,∀k∈l, the user with index Idl is the preselected orthogonal user of the lth multicast group.5. end**Output**: Orthogonal users among the different multicast groups.**Step 2: User preselection algorithm in each multicast group.****Input**: Orthogonal users among the different multicast groups, CSI.1. For l=1:L2.   For other users in the lth multicast group except the orthogonal user preselected in step 1, calculate the linear correlation value between each user and the preselected orthogonal user of the multicast group in turn, i.e., Ck,l=hk,lHhIdlhIdlHhIdlH22.3.   Based on the Ck,l of users in each multicast group, select top Ul,p−1 largest users, plus the orthogonal users in step 1 as the preselected users of each multicast group.4. end5. end**Output**: Preselected users for each multicast group.

After the user preselection, the joint user scheduling and hybrid beamforming design is for the preselected users. Therefore, the dimension of the LEO satellite communication system will be reduced, and the algorithm complexity will be reduced. Although the algorithm performance has a slight loss, compared with the decoupling design of user scheduling and beamforming, the performance is greatly improved. In conclusion, the joint user scheduling and hybrid beamforming design with the user preselection step is a better choice after balancing performance and complexity.

## 7. Solution of The Digital Beamforming Matrix and The Analog Beamforming Matrix

In this section, we aim to investigate the design of digital beamforming matrix $F_{BB}$ and analog beamforming matrix $F_{RF}$ in a hybrid beamformer. After obtaining W, we need to further solve $F_{BB}$ and $F_{RF}$. The solution method of $F_{BB}$ and $F_{RF}$ can be divided into two steps:The first step: we adopt the EVD algorithm to solve the hybrid beamforming matrix F from W.The second step: we propose the MM-AltOpt algorithm to obtain $F_{BB}$ and $F_{RF}$.

### 7.1. Solution of The Hybrid Beamforming Matrix

Before calculating $F_{BB}$ and $F_{RF}$, it is necessary to obtain the hybrid beamforming matrix F. For the solution of F, we can adopt the EVD algorithm based on the previously obtained optimization variable W. According to the relationship between W and F, i.e., $W_{l}≜f_{l}f_{l}^{H}$, the solution of F can be modeled as follows:(80)$$\underset{f_{l}}{\min}{\left\|W_{l}-f_{l}f_{l}^{H}\right\|}_{F}^{2},$$
where the hybrid beamforming vector $f_{l}$ can be given by
(81)$$f_{l}=\sqrt{v_{l}}u_{l},\forall l,$$
where $v_{l}$ is the maximum eigenvalue of the matrix $W_{l}$ and $u_{l}$ is the maximum eigenvector of the matrix $W_{l}$.

### 7.2. MM-AltOpt Algorithm: Solution of $F_{BB}$ and $F_{RF}$

According to $F=F_{RF}F_{BB}$ and $\left|{\left(F_{RF}\right)}_{i,j}\right|=1$, the solution of $F_{BB}$ and $F_{BB}$ can be modeled as a joint optimization problem with the power and constant modulus constraints, as follows:(82)$$P_{1}:\underset{F_{BB},F_{RF}}{\min}\;{\left\|F^{opt}-F_{RF}F_{BB}\right\|}_{F}^{2},$$
(83)$$s.t.\;C_{1}:F_{RF}\in F,$$
(84)$$C_{2}:{\left\|F_{RF}F_{BB}\right\|}_{F}^{2}=P_{T},$$
where $F=\left\{F_{RF}\in C^{N\times L}|{\left|F_{RF}{}_{\left(i,j\right)}\right|}^{2}=1,1\leq i\leq N,1\leq j\leq L\right\}$ represents the unit modulus constraint, which is determined by the phase shifter in UPA, and the constraint $C_{2}$ represents the power constraint.

It is worth noting that the problem $P_{1}$ is a matrix decomposition problem with the constant modulus constraint and the equality constraint. The objective function is a nonconvex function of variables $F_{BB}$ and $F_{RF}$, and the constraints $C_{1}$ and $C_{2}$ are also nonconvex. Meanwhile, it can be seen that when one of the two variables is given, the objective function is the convex function of the other variable. To solve the problem $P_{1}$, we invoke the alternating optimization algorithm. The alternating optimization algorithm can decompose the multivariable joint optimization problem into multiple subproblems according to the partial convexity of the problem $P_{1}$, and one of the variables can be iteratively solved by fixing the residual variables.

It should be noted that the nonconvexity of constraints is still a challenge. To this end, we first relax the constraint $C_{2}$, and then use the scale factor to adjust the digital beamforming matrix $F_{BB}$ to meet the power constraint. Then, for the solution of the analog beamforming matrix $F_{RF}$ with the unit modulus constraint, the MM algorithm is adopted [33].

### 7.2.1. Solution of The Analog Beamforming Matrix Based on The MM Algorithm

According to the solution process of the alternating optimization algorithm, we first solve the analog beamforming matrix $F_{RF}$ based on the digital beamforming matrix $F_{BB}$. Thus, the problem $P_{1}$ can be expressed as:(85)$$P_{2}:\underset{F_{RF}}{\min}\;{\left\|F-F_{RF}F_{BB}^{\left(n\right)}\right\|}_{F}^{2},$$
(86)$$s.t.\;C_{1}:F_{RF}\in F,$$
where $F_{BB}^{\left(n\right)}$ represents the estimated value of the digital beamforming matrix $F_{BB}$ at the nth iteration. Due to the unit module constraint of elements in $F_{RF}$, the problem $P_{2}$ is a nonconvex optimization problem.

According to the MM framework theory, the key step is constructing a surrogate function of the objective function in the optimization problem [39]. To construct the surrogate function, we decompose the matrix F by rows. According to the equivalence of the F-norm and $L_{2}$-norm of the vector, the problem $P_{2}$ can be rewritten as:(87)$$P_{3}:\;\underset{F_{RF}}{\min}\;\sum_{i=1}^{N}F_{i}^{H}F_{i}-2ℜ\left(F_{i}^{H}F_{BB}^{\left(n\right)}F_{RF,i}\right)+F_{RF,i}^{H}F_{BB}^{\left(n\right)}F_{BB}^{\left(n\right)}{}^{H}F_{RF,i},$$
(88)$$s.t.\;C_{1}:F_{RF}\in F,$$
where $F_{i}^{H}$ represents the ith row vector of the matrix F, and $F_{RF,i}^{H}$ represents the ith row vector of the matrix $F_{RF}$.

It should be noted that the third term $F_{RF,i}^{H}F_{BB}^{\left(n\right)}F_{BB}^{\left(n\right)}{}^{H}F_{RF,i}$ in (86) is a convex function term, which needs further conversion. According to the first-order Taylor expansion, $F_{RF,i}^{H}F_{BB}^{\left(n\right)}F_{BB}^{\left(n\right)}{}^{H}F_{RF,i}$ can be converted into:(89)$$\begin{array}{ll} F_{RF,i}^{H}F_{BB}^{\left(n\right)}F_{BB}^{\left(n\right)}{}^{H}F_{RF,i} & =F_{RF,i}^{\left(q\right)H}F_{BB}^{\left(n\right)}F_{BB}^{\left(n\right)}{}^{H}F_{RF,i}^{\left(q\right)}+2ℜ\left(F_{RF,i}^{\left(q\right)H}F_{BB}^{\left(n\right)}F_{BB}^{\left(n\right)}{}^{H}\left(F_{RF,i}-F_{RF,i}^{\left(q\right)}\right)\right)\\ & +{\left(F_{RF,i}-F_{RF,i}^{\left(q\right)}\right)}^{H}F_{BB}^{\left(n\right)}F_{BB}^{\left(n\right)}{}^{H}\left(F_{RF,i}-F_{RF,i}^{\left(q\right)}\right) \end{array}$$
where $F_{RF,i}^{\left(q\right)}$ represents the estimated value of $F_{RF,i}^{}$ at the qth iteration. According to the MM algorithm, the surrogate of $F_{RF,i}^{H}F_{BB}^{\left(n\right)}F_{BB}^{\left(n\right)}{}^{H}F_{RF,i}$ can be expressed as follows:(90)$$\begin{array}{ll} F_{RF,i}^{H}F_{BB}^{\left(n\right)}F_{BB}^{\left(n\right)}{}^{H}F_{RF,i} & \leq F_{RF,i}^{\left(q\right)H}F_{BB}^{\left(n\right)}F_{BB}^{\left(n\right)}{}^{H}F_{RF,i}^{\left(q\right)}+2ℜ\left(F_{RF,i}^{\left(q\right)H}F_{BB}^{\left(n\right)}F_{BB}^{\left(n\right)}{}^{H}\left(F_{RF,i}-F_{RF,i}^{\left(q\right)}\right)\right)\\ & +{\left(F_{RF,i}-F_{RF,i}^{\left(q\right)}\right)}^{H}X^{\left(n\right)}\left(F_{RF,i}-F_{RF,i}^{\left(q\right)}\right) \end{array}$$
where $X^{\left(n\right)}$ is a positive semidefinite matrix and satisfies the constraint $X^{\left(n\right)}\underline{≻}F_{BB}^{\left(n\right)}F_{BB}^{\left(n\right)}{}^{H}$; here, we let $X^{\left(n\right)}=\lambda_{\max}\left(F_{BB}^{\left(n\right)}F_{BB}^{\left(n\right)}{}^{H}\right)I$ and $\lambda_{\max}\left(F_{BB}^{\left(n\right)}F_{BB}^{\left(n\right)}{}^{H}\right)$ represents the maximum eigenvalue of the matrix $F_{BB}^{\left(n\right)}F_{BB}^{\left(n\right)}{}^{H}$. In conclusion, (89) can be further expressed as:(91)$$\begin{array}{ll} F_{RF,i}^{H}F_{BB}^{\left(n\right)}F_{BB}^{\left(n\right)}{}^{H}F_{RF,i} & \leq \lambda_{\max}\left(F_{BB}^{\left(n\right)}F_{BB}^{\left(n\right)}{}^{H}\right)F_{RF,i}^{H}F_{RF,i}+2ℜ\left(F_{RF,i}^{H}\left(F_{BB}^{\left(n\right)}F_{BB}^{\left(n\right)}{}^{H}-\lambda_{\max}\left(F_{BB}^{\left(n\right)}F_{BB}^{\left(n\right)}{}^{H}\right)I\right)F_{RF,i}\right)\\ & +F_{RF,i}^{\left(q\right)H}\left(\lambda_{\max}\left(F_{BB}^{\left(n\right)}F_{BB}^{\left(n\right)}{}^{H}\right)I-F_{BB}^{\left(n\right)}F_{BB}^{\left(n\right)}{}^{H}\right)F_{RF,i}^{} \end{array}$$

According to (90), the surrogate function of the objective function in the problem $P_{3}$ can be expressed as:


(92)
$$P_{3}:\underset{F_{RF}}{min}\;\overset{N}{\underset{i=1}{\sum }}F_{i}^{H}F_{i}-2ℜ\left(F_{i}^{H}F_{BB}^{\left(n\right)}F_{RF,i}\right)+F_{RF,i}^{H}F_{BB}^{\left(n\right)}F_{BB}^{\left(n\right)}{}^{H}F_{RF,i}\Rightarrow \;P_{4}:\underset{F_{RF}}{min}\;\overset{N}{\underset{i=1}{\sum }}F_{i}^{H}F_{i}-2ℜ\left(F_{i}^{H}F_{BB}^{\left(n\right)}F_{RF,i}\right)+\lambda {\left(F_{BB}^{\left(n\right)}F_{BB}^{\left(n\right)}{}^{H}\right)}_{RF,i}^{H}{{}_{RF,i}}_{max}\;+2ℜ\left(F_{RF,i}^{H}\left(F_{BB}^{\left(n\right)}F_{BB}^{\left(n\right)}{}^{H}-\lambda {\left(F_{BB}^{\left(n\right)}F_{BB}^{\left(n\right)}{}^{H}\right)}_{max}{\left(\right)}_{RF,i}\right){\left(\right)}_{RF,i}^{\left(q\right)H}\left(\lambda {\left(F_{BB}^{\left(n\right)}F_{BB}^{\left(n\right)}{}^{H}\right)}_{BB}^{\left(n\right)}{{}_{BB}^{\left(n\right)}}^{H}{}_{max}{\left(\right)}_{RF,i}\right)\right),$$


It is worth noting that the first and third terms of the objective function in the problem $P_{4}$ are constant terms, and the last term is independent of the variable $F_{RF,i}^{H}$. After ignoring the above three items, the problem $P_{4}$ can be converted to the following projection problem:(93)$$P_{5}:\;\underset{F_{RF}}{\min}\;\sum_{i=1}^{N}{\left\|F_{RF,i}-c_{i}^{\left(q\right)}\right\|}_{2}^{2},$$
(94)$$s.t.\;C_{1}:F_{RF}\in F,$$
where $c_{i}^{\left(q\right)}=F_{BB}^{\left(n\right)}F_{i}-\left(F_{BB}^{\left(n\right)}F_{BB}^{\left(n\right)}{}^{H}-\lambda_{\max}\left(F_{BB}^{\left(n\right)}F_{BB}^{\left(n\right)}{}^{H}\right)I\right)F_{RF,i}^{\left(q\right)H}$.

Therefore, the following closed form solution can be obtained for the problem $P_{5}$, which is reflected in line three of the **inner algorithm** in Algorithm 3:(95)$$F_{RF,i}^{}=e^{j\arg\left(c_{i}^{\left(q\right)}\right)},\forall i,$$
(96)$$F_{RF}^{}=e^{-j\arg\left(C^{\left(q\right)T}\right)},$$
where $C^{\left(q\right)}=F_{BB}^{\left(n\right)}F^{H}-\left(F_{BB}^{\left(n\right)}F_{BB}^{\left(n\right)}{}^{H}-\lambda_{\max}\left(F_{BB}^{\left(n\right)}F_{BB}^{\left(n\right)}{}^{H}\right)I\right)F_{RF}^{\left(q\right)H}$, which is reflected in line two of the **inner algorithm** in Algorithm 3.

### 7.2.2. Solution of The Digital Beamforming Matrix

Based on the analog beamforming matrix $F_{RF}$ obtained in the previous section, the solution problem of the digital beamforming matrix $F_{BB}$ can be modeled as follows:(97)$$P_{6}:\underset{F_{BB}}{\min}\;{\left\|F-F_{RF}^{\left(n\right)}F_{BB}\right\|}_{F}^{2},$$
(98)$$s.t.\;{\left\|F_{RF}^{\left(n\right)}F_{BB}\right\|}_{F}^{2}=P_{T},$$
where $F_{RF}^{\left(n\right)}$ represents the estimated value of the analog beamforming matrix $F_{RF}$ at the nth iteration. Due to the quadratic form and the convex equality constraint in the problem $P_{6}$, the problem $P_{6}$ is a nonconvex QCQP form problem.

One of the ways to solve the problem $P_{6}$ is to relax the equality constraint into the inequality constraint, and then convert the problem $P_{6}$ into a convex minimization problem, which can be solved with the CVX toolbox, but the complexity of this method is high. To this end, based on the fact that the hybrid beamforming matrix F satisfies the power constraint, i.e., ${\left\|F\right\|}_{F}^{2}=P_{T}$, the following closed form solution can be obtained for the problem $P_{6}$, which is reflected in line two of the **main**
**algorithm** in Algorithm 3:(99)$$F_{BB}={\left(F_{RF}^{\left(n\right)H}F_{RF}^{\left(n\right)}\right)}^{-1}F_{RF}^{\left(n\right)H}F,$$

In conclusion, the MM-AltOpt algorithm for solving FBB and FRF can be described as the main algorithm and the inner algorithm, as follows:
**Algorithm 3:** Design algorithm of the digital beamforming matrix and the analog beamforming matrix.**Main algorithm: MM-AltOpt algorithm.****Input**: Hybrid beamforming matrix F, initial: FRFn=0∈F, iteration index n=0, threshold ε3=10−3, the solution of the objective function of the problem P1 in the nth iteration is δn.1. while δn−δn−1≥ε32.   Based on FRFn, calculate FBBn+1 according to FBB=FRFnHFRFn−1FRFnHF.3.   Based on FBBn+1, calculate FRFn+1 according to the **inner algorithm**.4.   Set n=n+1.5. end**Output**: FRF,FBB, normalize FBB=PTFRFFBBFFBB.**Inner algorithm:**
**Algorithm for solving the analog beamforming matrix.****Input**: Hybrid beamforming matrix F, FBBn, FRFq=0∈F, iteration index q=0, threshold ε4=10−3, the solution of the objective function of the problem P5 in the qth iteration is δ1q.1. while δn−δn−1≥ε42.     Calculate Cq=FBBnFH−FBBnFBBnH−λmaxFBBnFBBnHIFRFqH.3.     Calculate FRF=e−jargCqT.4.     Set q=q+1.5. end**Output**: FRFn.

## 8. Results and Discussion

In this section, we evaluate the performance of the proposed joint user scheduling and hybrid beamforming design algorithm by numerical simulations. In the numerical simulations, we set the number of multicast groups to L=7, which cover 150 active users, and set the SINR constraint threshold of each multicast group to $SINR_{0}=1$. To facilitate analysis, we assume that the CSI errors of different multicast groups are the same, which are expressed as $\sigma_{f_{d,k,l}^{rsd}}^{2}=\sigma_{f_{d}^{rsd}}^{2}$ and $\sigma_{\phi_{\tau_{k,l}}}^{2}=\sigma_{\phi_{\tau }}^{2}$. The value of $P_{0}$ can be calculated by [31]. In addition, the system parameters used in the numerical simulations are shown in Table 1.

Figure 7 shows the convergence trajectory of the EE of the massive MIMO LEO satellite multigroup multicast communication system, versus the number of iterations for different CSI errors, different numbers of preselected users and different scheduling algorithms. In this simulation, two groups of channel errors are set according to Refs. [23,29], i.e., $\sigma_{f_{d}^{rsd}}^{2}=25$, $\sigma_{\phi_{\tau }}^{2}=10$ and $\sigma_{f_{d}^{rsd}}^{2}=20$, $\sigma_{\phi_{\tau }}^{2}=5$. In addition, we set $U_{s}=2$ and $P_{T}=50\;W$. Meanwhile, we set two different numbers of the preselected users, i.e., $U_{l,p}/U_{s}=2,U_{l,p}=4$ and $U_{l,p}/U_{s}=3,U_{l,p}=6$. It can be seen that the proposed robust algorithm has higher performance gain than the traditional nonrobust algorithm, which shows the effectiveness of the robust algorithm. Meanwhile, it can be seen that when $\sigma_{f_{d}^{rsd}}^{2}=25$, $\sigma_{\phi_{\tau }}^{2}=10$ and $\sigma_{f_{d}^{rsd}}^{2}=20$, and $\sigma_{\phi_{\tau }}^{2}=5$, the EE performance gain of the proposed robust algorithm is improved by 9.8% and 6.7%, respectively, compared with the traditional nonrobust algorithm. The system EE of the proposed joint user scheduling and hybrid beamforming design algorithm is higher than that of the decoupling design algorithm, and the more preselected users, the higher performance improvement.

Figure 8 compares the EE performance of the proposed algorithm and the traditional algorithm under different system parameters, versus different transmission power thresholds $P_{T}$. It can be seen that with the increase in transmission power, the EE performance shows a trend of first rising and then falling. The reason is that the growth rate of the system rate is lower than that of the power consumption. Meanwhile, we can see that the EE of the joint user scheduling and hybrid beamforming design algorithm is higher than that of the decoupling design algorithm versus different transmission power thresholds $P_{T}$. Under the conditions of $\sigma_{f_{d}^{rsd}}^{2}=25$, $\sigma_{\phi_{\tau }}^{2}=10$ and $P_{T}=15\;W$, when $U_{l,p}/U_{s}=2,U_{l,p}=4$ and $U_{l,p}/U_{s}=3,U_{l,p}=6$, the EE performance gain of the proposed joint design algorithm is 28.41% and 45.19% higher than that of the traditional decoupling design algorithm.

Figure 9 indicates the change trend of the SE of the massive MIMO LEO satellite multigroup multicast communication system versus different transmission power thresholds $P_{T}$. It can be seen that the SE increases with the increase in the transmission power. Meanwhile, we can see that the SE of the joint user scheduling and hybrid beamforming design algorithm is higher than that of the decoupling design algorithm. In addition, the more preselected users, the higher the system SE. Compared with the traditional algorithm, the proposed robust joint design algorithm can obtain higher system SE at the same transmission power, and thus can improve the system EE. Under the conditions of $\sigma_{f_{d}^{rsd}}^{2}=25$, $\sigma_{\phi_{\tau }}^{2}=10$ and $P_{T}=30\;W$, when $U_{l,p}/U_{s}=2,U_{l,p}=4$ and $U_{l,p}/U_{s}=3,U_{l,p}=6$, with the improvement of system SE performance, the EE performance gain of the proposed joint design algorithm is 26.16% and 37.85% higher than that of the traditional decoupling design algorithm.

Figure 10 shows the SE comparison of different multicast groups. It can be seen that the SE of each multicast group of the proposed robust algorithm is higher than that of the nonrobust algorithm. Meanwhile, with the increase in the number of preselected users, the diversity of users increases, and the performance of the proposed joint user scheduling and hybrid beamforming design algorithm also improves. This is because with the increase in the number of preselected users, the range of users that can be scheduled and selected increases. By scheduling different users in each multicast group, the SE can be further improved. Meanwhile, with the improvement of the system SE, the system EE performance gain also increases, which verifies the effectiveness of the joint user scheduling and hybrid beamforming design algorithm.

Figure 11 shows the change trajectory of the system EE and SE versus the different $U_{s}$. It can be seen that with the increase in $U_{s}$, the system EE and SE show a downward trend. This is because the communication rate of each multicast group is constrained by the user with the worst SINR in the multicast group. With the increase in $U_{s}$, if the users’ channel vectors in the multicast group remain collinear, the EE and SE would remain unchanged. However, according to the rules of the user preselection and scheduling, with the increase in $U_{s}$, the collinearity among users in the multicast group would decrease, which can result in the increase in interference and the decrease in the worst SINR in each multicast group. In other words, with the decrease in $U_{s}$, the users’ SINR will be improved. Therefore, with the improvement of SINR, the system EE performance gain also increases, as shown in Figure 11a. Under the condition of $U_{l,p}/U_{s}=2,U_{l,p}=4$, when $U_{s}=2$, the EE performance gain is 21.05% higher than when $U_{s}=4$.

Figure 12 shows the performance comparison of different algorithms for solving $F_{BB}$ and $F_{RF}$. In this simulation, we set three comparison algorithms, i.e., the optimal design algorithm, the alternating minimization algorithm based on the phase extraction (PE-Altmin) algorithm and the orthogonal matching pursuit (OMP) algorithm. The optimal design algorithm refers to the numerical simulation result of the hybrid beamforming matrix F. It can be seen that the system performance of the MM-AltOpt algorithm is slightly lower than that of the optimal design algorithm. Meanwhile, in Figure 12a, when $P_{T}=10\;W$, we can see that the system EE performance gain of the proposed MM-AltOpt algorithm is improved by about 2% and 5%, respectively, compared with the PE-Altmin algorithm and the OMP algorithm. In addition, from the perspective of algorithm complexity, the complexities of the MM-AltOpt algorithm, PE-Altmin algorithm and OMP algorithm are $O\left(I_{MM}\left(N^{3}+I_{Inner}2NL^{2}+NL\right)\right)$, $O\left(I_{PE}\left(N^{3}+L^{3}+NL\right)\right)$ and $O\left(I_{OMP}\left(L^{4}N+L^{2}+N^{2}L^{2}+2L^{3}\right)\right)$, respectively, where $I_{MM}$, $I_{Inner}$, $I_{PE}$ and $I_{OMP}$ are the number of iterations of the corresponding algorithm. In conclusion, we can see that the complexity of the proposed MM-AltOpt algorithm is close to that of the other two algorithms, however, the system EE performance is higher, which can verify the effectiveness of the proposed algorithm.

## 9. Conclusions

In this paper, we investigated the robust joint user scheduling and hybrid beamforming design scheme for the massive MIMO LEO satellite multigroup multicast communication system. The scheme design considered the limited transmission power of the LEO satellite and the requirement of QoS and analyzed the influence of residual Doppler shift and phase disturbance on CSI errors. On this basis, taking the system EE as the optimization objective, we focused on the robust joint user scheduling and hybrid beamforming design. To reduce the complexity of the algorithm, we proposed the user preselection step, which can significantly reduce the system complexity while ensuring the system performance. For the nonconvex problem of the objective function, we adopted the CCCP framework after transforming the optimization problem into the DC programming problem. For the rank-one constraint, we proposed the penalty iterative algorithm. Finally, to obtain the digital and analog beamforming matrices, we adopted the MM-AltOpt algorithm.

Numerical results indicated that the proposed algorithm can effectively improve the system EE. The EE performance gain of the proposed robust algorithm was improved by nearly 10% compared with the traditional nonrobust algorithm. Meanwhile, the EE performance gain of the proposed joint user scheduling and hybrid beamforming design algorithm was improved by nearly 40% compared with the traditional decoupling design algorithm. In conclusion, the robust joint user scheduling and hybrid beamforming design algorithm proposed in this paper can significantly improve the system EE performance.

## Figures and Tables

**Figure 1 sensors-22-06858-f001:**
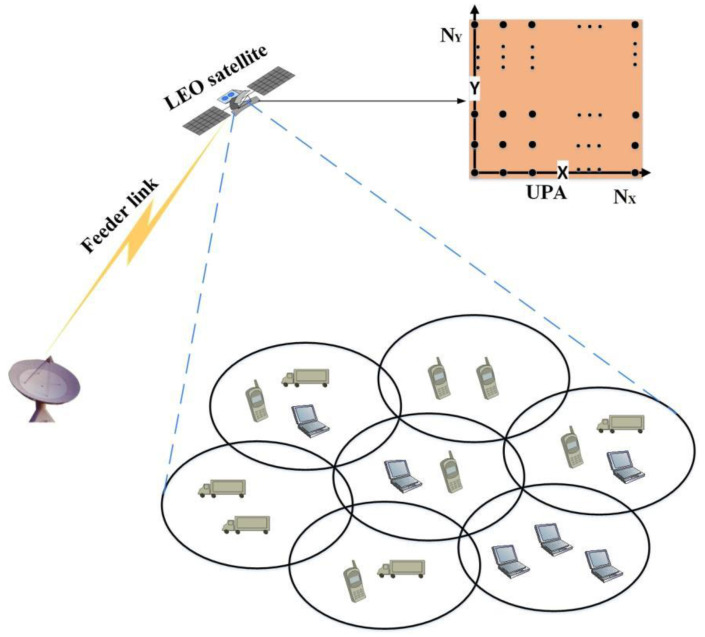
Transmission model of the massive MIMO LEO satellite multigroup multicast communication system.

**Figure 2 sensors-22-06858-f002:**
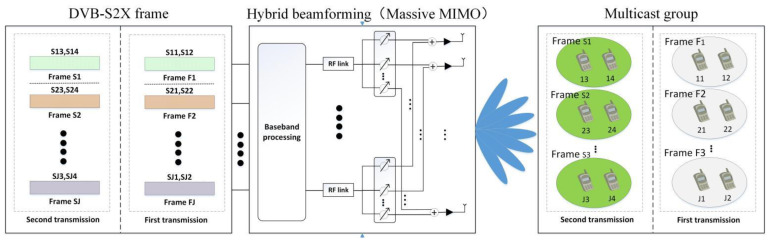
Service process of the LEO communication satellite multigroup multicast transmission based on the DVB-S2X.

**Figure 3 sensors-22-06858-f003:**
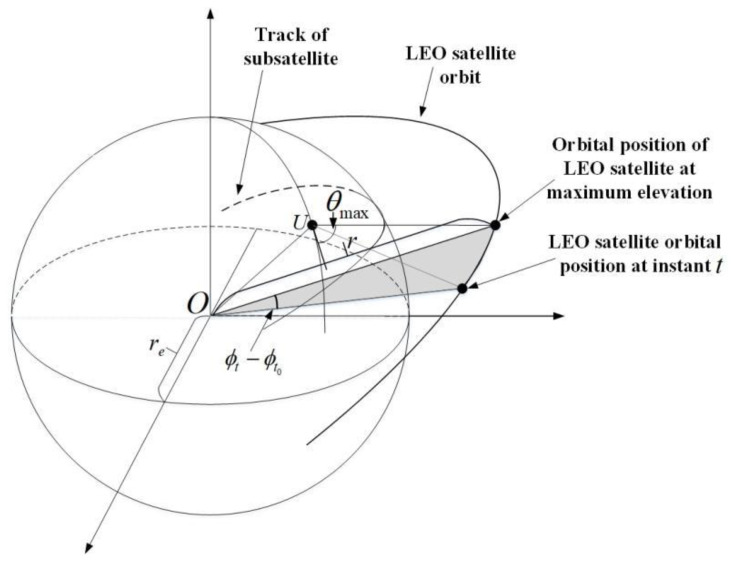
Geometric diagram of LEO satellite’s orbital motion relative to the user terminal.

**Figure 4 sensors-22-06858-f004:**
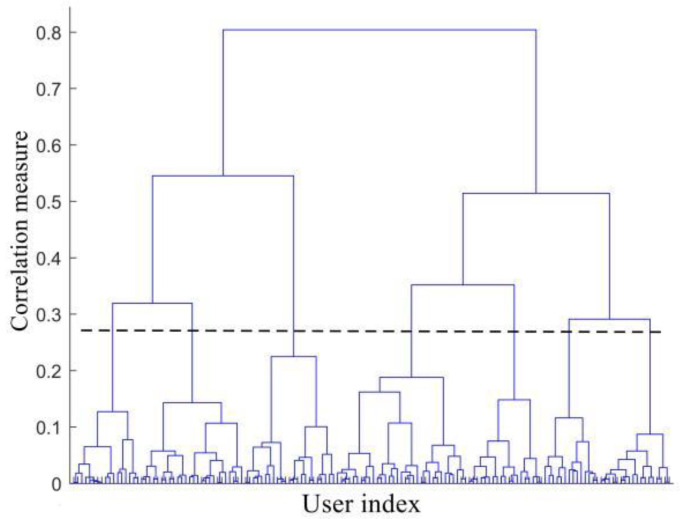
Schematic diagram of the hierarchical clustering algorithm.

**Figure 5 sensors-22-06858-f005:**
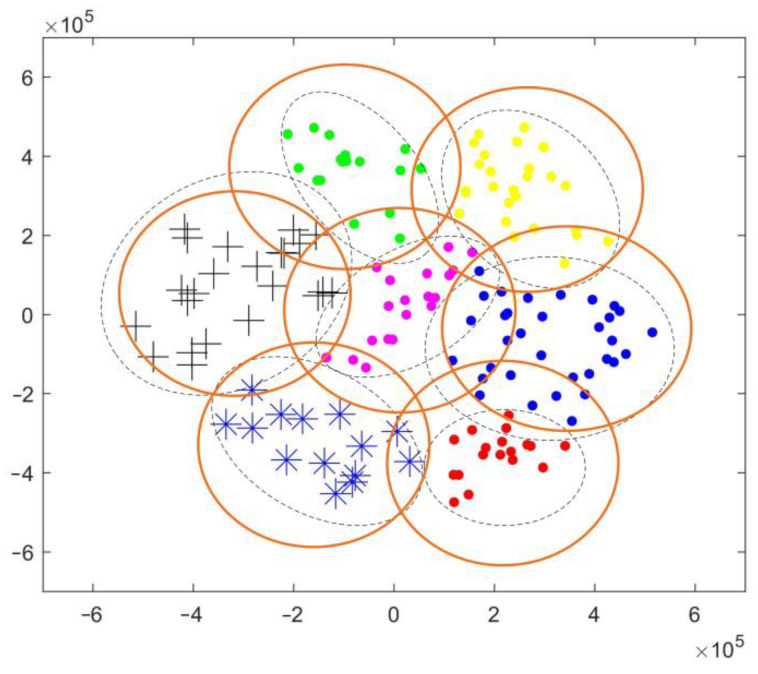
Schematic diagram of multibeam coverage with 7 multicast groups.

**Figure 6 sensors-22-06858-f006:**
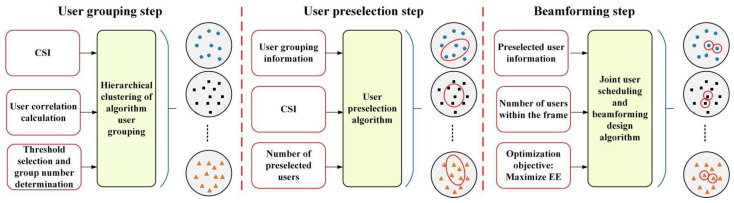
Design process of joint user scheduling and beamforming with the user preselection.

**Figure 7 sensors-22-06858-f007:**
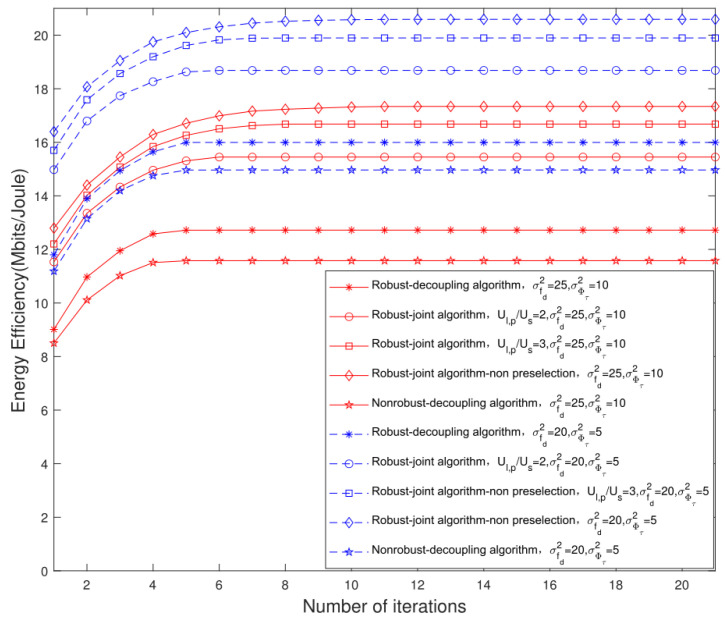
Convergence trajectory of system EE relative to different CSI errors, different number of preselected users and different scheduling algorithms.

**Figure 8 sensors-22-06858-f008:**
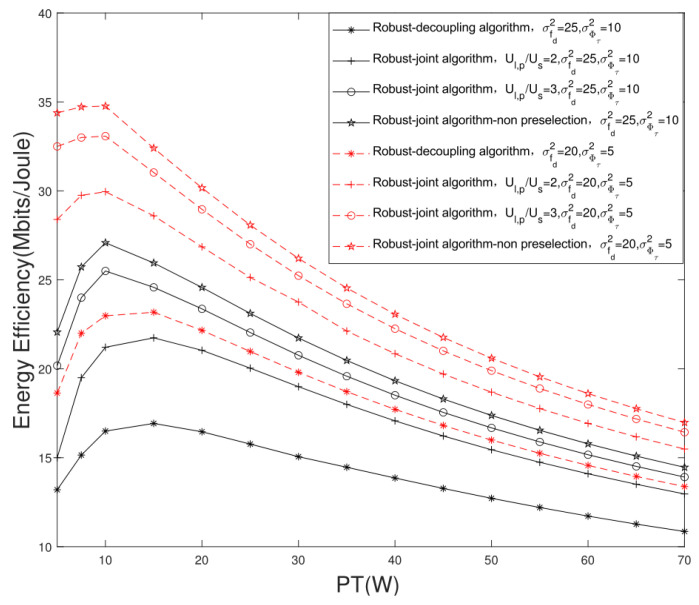
Comparison of system EE of different algorithms with different transmission power thresholds $P_{T}$.

**Figure 9 sensors-22-06858-f009:**
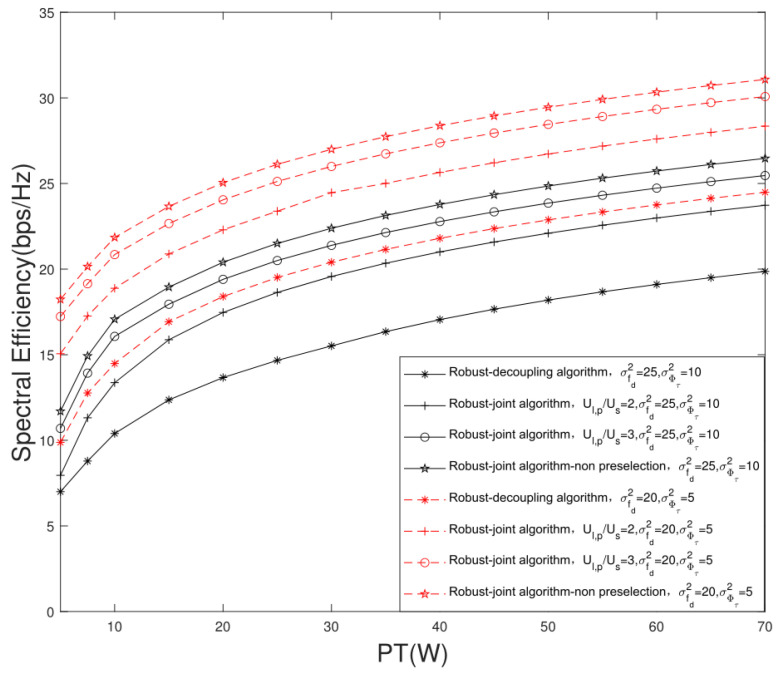
Change trajectory of system SE versus different transmission power thresholds $P_{T}$.

**Figure 10 sensors-22-06858-f010:**
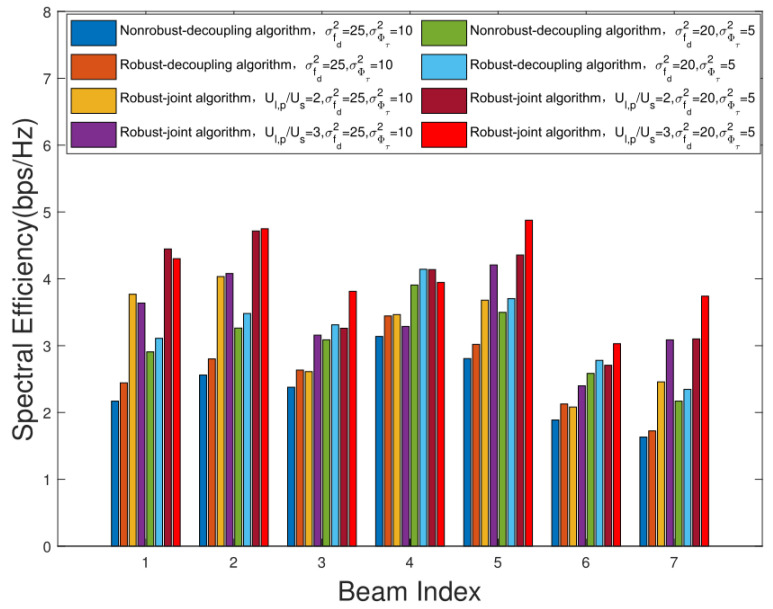
Comparison of SE of different multicast groups.

**Figure 11 sensors-22-06858-f011:**
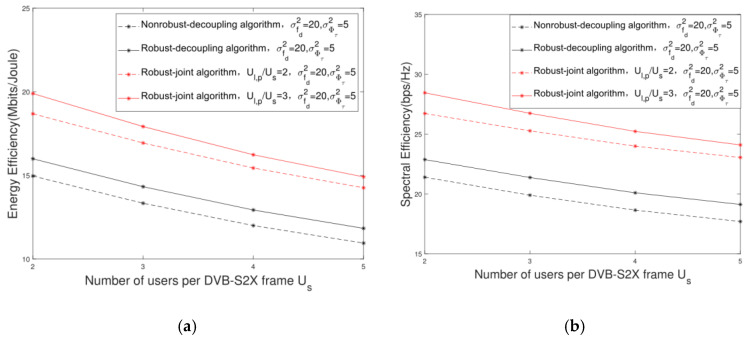
Change trajectory of system performance versus different $U_{s}$. (**a**) Change trajectory of system EE versus different $U_{s}$; (**b**) change trajectory of system SE versus different $U_{s}$.

**Figure 12 sensors-22-06858-f012:**
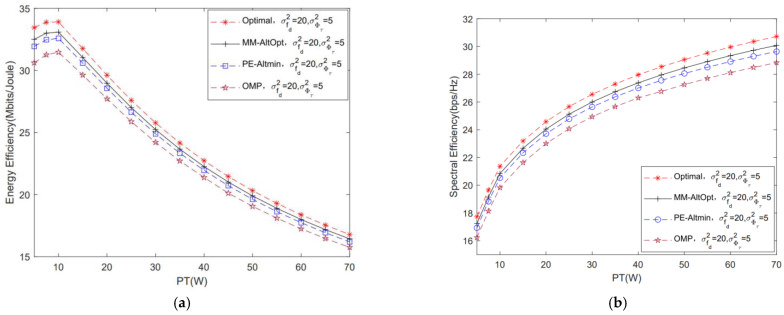
Comparison of system performance of different algorithms for solving $F_{BB}$ and $F_{RF}$. (**a**) Comparison of system EE of different algorithms for solving $F_{BB}$ and $F_{RF}$; (**b**) comparison of system SE of different algorithms for solving $F_{BB}$ and $F_{RF}$.

**Table 1 sensors-22-06858-t001:** Simulation parameters.

Parameters	Values	Parameters	Values
N	8×8=64	κ	$1.38\times 10^{-23}J\cdot K^{-1}$
L	7	$P_{0}$	21.5 W
$\kappa_{k,l}$	10	T	300 K
Bandwidth	50 MHz	$G_{leo}$	3 dB
Orbit altitude	1000 km	$G_{ut}$	3 dB
Beam radius	250 km	f	20 GHz
$LP_{at}$	0.017 dB	K	150

## Data Availability

The data presented in this study are available in the manuscript.

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
