# Peer review of "Joint User Scheduling and Hybrid Beamforming Design for Massive MIMO LEO Satellite Multigroup Multicast Communication Systems"

_sensors, 2022, doi:10.3390/s22186858_

Round 1

Reviewer 1 Report

1. Is there any atmospheric effect have been considered in this work, especially in propagation delay analysis? If no, where is the exact specific location of the users?

2. What is the spacing between Array antenna at the satellite? Did you considered the mutual effect that may be cause by the not enough spacing between antenna element?

3. This paper mentioned about non accurate CSI of the MIMO channel. What is the most effect cause by the CSI errors towards QoS?

4. Did you considered the effect of signal reflections at the user side? If no, what is the limitation of these analysis work in term of signal propagation mechanism?

Reviewer 2 Report

The authors present  a Joint User Scheduling and Hybrid Beamforming Design for Massive MIMO LEO Satellite Multigroup Multicast Communication Systems. It is a detailed paper with an interesting topic.  Some comments in order to improve the manuscript:   - All acronyms must be defined when first used in the text; - Some equations are very difficult to read without an important zoom, I suggest to cut these equations in parts; - Curves legends in Figs 7, 8, 9 and 11 must appear at the right or left of the figures an not into the figures; - References need to be reviewed, more consistent references are requested.
